# LeanQuant: Accurate and Scalable Large Language Model Quantization with Loss-error-aware Grid

**Tianyi Zhang**
Dept. of Computer Science, Rice University
xMAD.ai
Houston, TX
tz21@rice.edu

**Anshumali Shrivastava**
Dept. of Computer Science, Rice University
xMAD.ai
ThirdAI Corp.
Ken Kennedy Institute
Houston, TX
anshumali@rice.edu

## Abstract

Large language models (LLMs) have shown immense potential across various domains, but their high memory requirements and inference costs remain critical challenges for deployment. Post-training quantization (PTQ) has emerged as a promising technique to reduce memory requirements and decoding latency. However, recent accurate quantization methods often depend on specialized computations or custom data formats to achieve better model quality, which limits their compatibility with popular frameworks, as they require dedicated inference kernels tailored to specific hardware and software platforms, hindering wider adoption. Furthermore, many competitive methods have high resource requirements and computational overhead for quantizing models, making it challenging to scale them to hundreds of billions of parameters. In response to these challenges, we propose LeanQuant (Loss-error-aware network Quantization), a novel quantization method that is accurate, versatile, and scalable. In the existing popular iterative loss-error-based quantization framework, we identify a critical limitation in prior methods: the min-max affine quantization grid fails to preserve model quality due to outliers in inverse Hessian diagonals. To overcome this fundamental issue, we propose learning loss-error-aware grids, instead of using non-adaptive min-max affine grids. Our approach not only produces quantized models that are more accurate but also generalizes to a wider range of quantization types, including affine and non-uniform quantization, enhancing compatibility with more frameworks. Extensive experiments with recent LLMs demonstrate that LeanQuant is highly *accurate*, comparing favorably against competitive baselines in model quality, and *scalable*, achieving very accurate quantization of Llama-3.1 405B, one of the largest open-source LLMs to date, using two Quadro RTX 8000-48GB GPUs in 21 hours. Our code is available at https://github.com/LeanModels/LeanQuant.

## 1 Introduction

Large language models (LLMs) have demonstrated impressive reasoning (Wei et al., 2022) and problem solving abilities (Kojima et al., 2022), and have shown the potential to bring transformative changes to various fields such as law (Kaddour et al., 2023), education (Kasneci et al., 2023), and medicine (Thirunavukarasu et al., 2023). However, deploying LLMs in a cost-effective manner presents significant challenges due to their substantial memory and computational demands (Chen et al., 2023), which hinders the accessibility and democratization of artificial intelligence (AI) (Kaddour et al., 2023).

Post-training quantization (PTQ) (Krishnamoorthi, 2018) is a promising technique for reducing the memory footprint of model inference by lowering the precision of a pre-trained model's parameters and storing them in a compact, low-bit-width format. PTQ offers the additional benefit of reducing

the decoding latency of LLMs by reducing memory reads, since LLM inference is often bottle-necked by memory bandwidth (Kim et al., 2023). Although quantization causes a certain amount of precision loss in the parameters, the model quality can be reasonably preserved even in lower bit widths (Frantar et al., 2022; Chee et al., 2024). For many tasks, a quantized model is preferred over a full model due to its better size-accuracy trade-off (Dettmers & Zettlemoyer, 2023). As open-source foundational models continue to scale up in size (Dubey et al., 2024), accurate and efficient quantization becomes essential for making AI accessible to a wider audience. For instance, serving Llama-3.1 405B (Dubey et al., 2024) with its original 16-bit weights requires a cluster of two nodes, each equipped with 8×80GB GPUs. In contrast, a 4-bit quantized version can be deployed on a single node with 8×48GB GPUs, eliminating inter-node communication overhead.

**Challenges of Deploying Quantized Models** One of the biggest challenges of successful deployment of quantized models is implementing optimized kernels for quantized GEMM (general matrix multiply) that are tailored to various hardware platforms and software frameworks. In order to accelerate inference of quantized models, fused kernels, which fuse dequantization and matrix multiplication in the same subroutine, have to be implemented and tuned for the specific hardware accelerator. These kernels require specialized designs and tunings for different hardware accelerators to be fully optimized (Park et al., 2022). Recent quantization algorithms have chosen to employ specialized computations or custom data formats to reduce the impact of quantization on model quality, but they require more sophisticated kernel designs for efficient inference. For example, AQLM (Egiazarian et al., 2024) and QUIP# (Tseng et al., 2024) perform dequantization through look-ups from multi-dimensional or multi-bit codebooks, and Dotzel et al. (2024) proposed new data types such as Student Float to reduce quantization errors. While these approaches demonstrate promising results, their reliance on specialized operations and data formats can hinder their widespread adoption due to the need for optimized inference kernels for each hardware platform and software framework. For example, llama.cpp (Gerganov, 2023), a popular LLM inference engine that supports mobile devices, only supports affine and non-uniform quantization formats. Consequently, instead of focusing on developing better quantization methods with specialized operations, it may be more worthwhile to investigate improving the accuracy of existing widely adopted quantization formats, such as affine integer quantization and non-uniform quantization, which are supported by popular deep learning libraries (Paszke et al., 2019) and deployment frameworks (Kwon et al., 2023).

**Scalability Challenges of Accurate Quantization** To improve the quality of quantized models, existing approaches often incur higher computational overhead and require more hardware resources. As foundational models scale up in size (Hoffmann et al., 2022), these quantization approaches may struggle to scale to very large models, such as Llama-3.1 405B (405 billion parameters) (Dubey et al., 2024). For instance, LLM-QAT (Liu et al., 2023) uses 100K samples of training data and hundreds of GPU-hours to recover the performance of a quantized LLaMA-13B model (Touvron et al., 2023a). For AQLM (Egiazarian et al., 2024), the time needed for quantizing a 7B to 70B LLM ranges from 1 to 14 days of an A100-80GB GPU. For SqueezeLLM (Kim et al., 2023), due to its use of the gradients of model parameters, quantizing a 70B LLM requires at least 240GB of total GPU memory. Given the significant hardware resources and lengthy optimization times of these quantization approaches, developing accurate yet efficient methods is crucial for ensuring accessibility of larger foundational models.

**Our Proposal** In this work, we propose LeanQuant, an accurate, versatile, and scalable quantization approach. We build upon the iterative loss-error-based quantization framework (Frantar & Alistarh, 2022; Frantar et al., 2022) and identify one of the biggest limitations of such methods: the min-max affine quantization grid introduces high loss errors due to the existence of outliers in the inverse Hessian diagonals. We introduce techniques for learning loss-error-aware quantization grids, which mitigate this issue and greatly improve the quality of quantized models. We empirically demonstrate that LeanQuant compares favorably against competitive baselines in the 4/3/2-bit regions. Our approach is versatile, able to generalize to multiple commonly used quantization formats, such as affine and non-uniform quantization, allowing our quantized models to be directly compatible with existing highly optimized inference kernels (Frantar et al., 2024; Park et al., 2022) for maximum accessibility. Furthermore, our method is scalable and efficient. By designing and implementing a fused GPU kernel for LeanQuant grid learning, we achieve the accurate quantization of LLMs up to 123B in size using a single L40s-48GB GPU in 4 hours, and Llama-3.1 405B using 2 Quadro RTX 8000-48GB GPUs in 21 hours.

## 2 BACKGROUND

In this section, we introduce the relevant background for our proposal including quantization grids and iterative loss-error-based quantization.

### 2.1 QUANTIZATION GRID

Quantization enables model compression by representing full-precision floating-point parameters with a limited set of grid points on a quantization grid. The number of available grid points is determined by the bit width: a $b$-bit code allows for $2^b$ distinct values, meaning 2-bit quantization results in four grid points. A visual explanation of quantization grid can be found in Appendix A. The placement of these points is crucial, as inaccurate placements can degrade model quality. To mitigate this, various quantization grids—such as affine, non-uniform, and normal—have been proposed. We provide an overview of these approaches below.

**Affine Grid** In an affine quantization grid (Krishnamoorthi, 2018), the grid points are evenly spaced between the minimum and maximum of a set of weights. To achieve finer quantization precision, the network's weights are divided into groups (e.g., every 128 contiguous parameters). In min-max asymmetric affine quantization, each weight group is associated with a scaling factor $S$ and a zero-point $Z$ (with $Z$ omitted in the symmetric case). The $i$-th weight $w_i$ in a group $\mathbf{w}$ is quantized to a $b$-bit integer $w_i^{\mathbf{int}}$ as follows:

$$w_i^{\mathbf{int}} = \text{clip}(\lfloor \frac{w_i}{S} \rceil + Z, 0, 2^b - 1), \text{where } S = \frac{\max(\mathbf{w}) - \min(\mathbf{w})}{2^b - 1} \text{ and } Z = -\lfloor \frac{\min(\mathbf{w})}{S} \rceil$$

$$\text{quant}_{aff}(w_i, S, Z) = (w_i^{\mathbf{int}} - Z)S$$

where $\lfloor \cdot \rceil$ denotes rounding, $\text{clip}(\cdot)$ ensures the value remains within the $b$-bit integer range, and $\text{quant}_{aff}(w_i, S, Z)$ is the quantized value of $w_i$.

**Non-uniform Grid** The grid points on a non-uniform grid are placed in a non-equidistant manner (Li et al., 2019). The motivation behind non-uniform quantization is to allow for finer precision in regions where model parameters are more concentrated or sensitive. Each row in a weight matrix has a distinct set of non-uniform grid points $\mathcal{G}$, where $|\mathcal{G}| = 2^b$ for $b$-bit quantization. The weight $w_i$ is quantized to the nearest grid point in $\mathcal{G}$ as follows,

$$\text{quant}_{nu}(w_i, \mathcal{G}) = \underset{g \in \mathcal{G}}{\arg\min} |g - w_i|$$

**Other Grid Types** Previous works have observed that the distribution of LLM parameters often resembles Normal or Student T's Distribution. Consequently, grid types such as NormalFloat (Dettmers et al., 2024) and Student Float (Dotzel et al., 2024) have been proposed, which align grid points with quantiles of these distributions. Our proposed method can be extended to support them.

### 2.2 ITERATIVE LOSS-ERROR-BASED QUANTIZATION

Iterative loss-error-based quantization (Frantar & Alistarh, 2022) is a promising framework for quantizing deep neural networks to low bit widths while preserving model quality. In particular, Optimal Brain Quantization (OBQ) (Frantar & Alistarh, 2022), which is based on the seminal works by LeCun et al. (1989) and Hassibi et al. (1993), aims to minimize the impact of weight perturbations introduced by parameter quantization on the network's task loss. Let $\mathcal{L}(\mathbf{w}_\mathcal{N})$ be the task loss of a network $\mathcal{N}$ evaluated at its weights $\mathbf{w}_\mathcal{N}$ (flattened to a vector). Then, the OBQ objective is to minimize the loss error $\epsilon$, which is defined as

$$\epsilon = \mathcal{L}(\mathbf{w}_\mathcal{N} + \boldsymbol{\delta}_\mathcal{N}) - \mathcal{L}(\mathbf{w}_\mathcal{N})$$

where $\boldsymbol{\delta}_\mathcal{N}$ is the weight perturbation introduced by quantization. The loss error $\epsilon$ can be approximated with a Taylor series (LeCun et al., 1989) as

$$\epsilon = \underbrace{(\frac{\partial \mathcal{L}}{\partial \mathbf{w}_\mathcal{N}})^\top \boldsymbol{\delta}_\mathcal{N}}_{\text{negligible}} + \frac{1}{2} \boldsymbol{\delta}_\mathcal{N}^\top \frac{\partial^2 \mathcal{L}}{\partial \mathbf{w}_\mathcal{N}^2} \boldsymbol{\delta}_\mathcal{N} + \underbrace{O(\|\boldsymbol{\delta}_\mathcal{N}\|^3)}_{\text{negligible}}$$

where the first term is omitted due to $\frac{\partial \mathcal{L}}{\partial \mathbf{w}_{\mathcal{N}}} \approx \mathbf{0}$ in a converged network, and the third and higher terms can be ignored due to small norms. Computing the exact Hessian $\mathbf{H} = \frac{\partial^2 \mathcal{L}}{\partial \mathbf{w}_{\mathcal{N}}^2}$ in a deep network is difficult, hence OBQ leverages an approximation of loss error proposed by Nagel et al. (2020),

$$\mathbb{E}(\epsilon) \approx \sum_{\mathbf{W} \in \mathcal{N}} \left\| \mathbf{W}\mathbf{X} - \hat{\mathbf{W}}\mathbf{X} \right\|_F^2$$

where $\mathbf{W}, \hat{\mathbf{W}}, \mathbf{X}$ are the weight matrix, quantized weight matrix, and the input matrix to a linear layer in the network $\mathcal{N}$. As a result, the OBQ objective can be decomposed into layer-wise independent convex problems,

$$\arg\min_{\hat{\mathbf{w}}} \|\mathbf{W}\mathbf{X} - \hat{\mathbf{W}}\mathbf{X}\|_F^2 \tag{1}$$

which can be further decomposed into row-wise independent problems, since Equation 1 can be written as a sum of squares over the rows of $\mathbf{W}$.

OBQ employs an iterative quantization approach, in which a single weight in a row $\mathbf{w}$ is quantized in each step, and then the remaining not-yet-quantized weights in the same row are updated to compensate for the introduced error. Given the constraint that the parameter $w_i$, indexed by $i$ in row $\mathbf{w}$, is being quantized, the optimal weight perturbation $\boldsymbol{\delta}$ to the remaining weights can be solved with the following Lagrangian,

$$L(\boldsymbol{\delta}, \lambda) = \frac{1}{2}\boldsymbol{\delta}^\top \mathbf{H} \boldsymbol{\delta} + \lambda\left(\mathbf{e}_i^\top \boldsymbol{\delta} - \big(\mathrm{quant}(w_i) - w_i\big)\right) \tag{2}$$

where $e_i$ is the $i$-th standard basis vector and $\mathbf{H} = 2\mathbf{X}\mathbf{X}^\top$ is the Hessian from Equation 1 (computed on a small sample of input data). Solving Equation 2 yields the optimal weight perturbation $\boldsymbol{\delta}_i$ and loss error $\epsilon_i$ after quantizing $w_i$,

$$\boldsymbol{\delta}_i = \frac{\mathrm{quant}(w_i) - w_i}{\mathbf{H}_{i,i}^{-1}} \mathbf{H}_{:,i}^{-1}, \quad \epsilon_i = \frac{1}{2}\frac{\big(\mathrm{quant}(w_i) - w_i\big)^2}{\mathbf{H}_{i,i}^{-1}} \tag{3}$$

where $\mathbf{H}_{i,i}^{-1}$ and $\mathbf{H}_{:,i}^{-1}$ denotes the $i$-th diagonal entry and the $i$-th column of the inverse Hessian, respectively.

The loss error $\epsilon_i$ quantifies the degradation in model quality caused by quantizing parameter $w_i$ and is always non-negative. OBQ leverages $\epsilon_i$ as a heuristic for greedy optimization. In each iteration, OBQ computes $\epsilon$ for all weights in a row and greedily selects the parameter $w_i$ with the smallest $\epsilon_i$ for quantization. The selected parameter is then rounded to the nearest value on the quantization grid, and the remaining weights are updated as $\mathbf{w} \leftarrow \mathbf{w} - \boldsymbol{\delta}_i$. This iterative process continues until all weights are quantized.

**Scaling to Billion-Parameter LLMs Using Cholesky and Dampening** OBQ produces accurate post-training quantized models for million-parameter networks, but fails to scale to billion-parameter LLMs due to two primary reasons: the inefficient time complexity and the accumulation of numerical inaccuracies during updates. To improve its computational efficiency, Frantar et al. (2022) propose to quantize the weights in a fixed non-greedy order for all rows, and keep the weight updates within a block of $B$ columns at a time. To prevent model quality collapse from the accumulation of numerical inaccuracies by repeated weight updates, Frantar et al. (2022) propose to apply a mild dampening (1% of the average diagonals) to the diagonal Hessian and leverage a Cholesky decomposition to compute the inverse Hessian $\mathbf{H}^{-1}$. The resulting algorithm is GPTQ, which can efficiently quantize billion-parameter LLMs.

## 3 METHODOLOGY

In this section, we introduce our proposed approach Loss-error-aware network Quantization (Lean-Quant), for accurately and efficiently quantizing LLMs.

### 3.1 REVISITING THE LOSS ERROR

To motivate our proposed approach, we first revisit the loss error $\epsilon_i$ in Equation 3, which approximates the (detrimental) increase in the network's task loss, introduced by quantizing weight $w_i$.

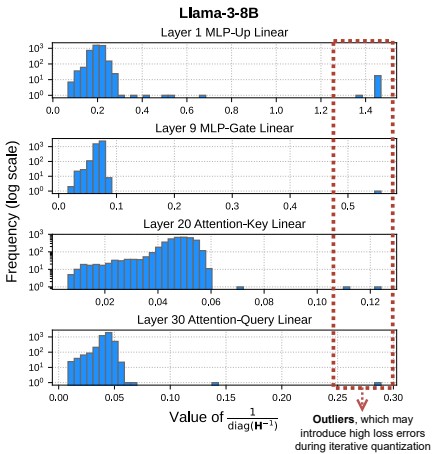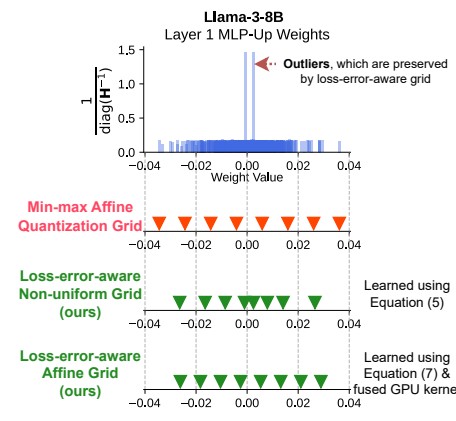

Figure 1: (Left) The empirical distributions of inverse Hessian diagonals, computed on 262K tokens from the C4 dataset for the Llama-3-8B model, contain outliers that can cause high loss errors. (Right) Our proposed loss-error-aware non-uniform and affine grids better preserve the quantized precision of outliers, leading to more accurate quantized models.

This error $\epsilon_i$ has been used as a heuristic in multiple previous works (LeCun et al., 1989; Hassibi et al., 1993; Singh & Alistarh, 2020; Frantar & Alistarh, 2022) for choosing the next best weight $i$ to prune or quantize. It has been shown to be a highly informative metric for measuring the impact of quantization.

By examining Equation 3, one finds that the loss error $\epsilon_i$ is proportional to the square of weight quantization error and inversely proportional to the diagonal entry of the inverse Hessian, i.e.,

$$\epsilon_i \propto \big(\mathrm{quant}(w_i) - w_i\big)^2 \ \text{ and } \ \epsilon_i \propto \frac{1}{\mathbf{H}_{i,i}^{-1}} \tag{4}$$

Hence, we further examine the empirical distribution of $\frac{1}{\mathrm{diag}(\mathbf{H}^{-1})}$, which is proportional to $\epsilon$, the loss error of an entire row. We obtain the empirical distributions on layers of Llama-3-8B (Dubey et al., 2024) with 128 sequences of length 2048 tokens from the C4 dataset (Raffel et al., 2020), and compute the inverse Hessian as $\mathbf{H}^{-1} = (2\mathbf{X}\mathbf{X}^\top)^{-1}$ where $\mathbf{X}$ is the layer input matrix. As shown in Figure 1, The majority of the inverse diagonals are concentrated in low-magnitude regions, with a few outliers having high magnitudes. Quantizing the weights corresponding to these outliers can lead to high loss errors if these weights are not well-aligned with the quantization grid points. Preserving the quantized precision of the weights corresponding to these inverse-diagonal outliers is especially important because the loss error increases quadratically with their quantization error (Equation 4). Iterative loss-error-based quantization approaches (OBQ, GPTQ, etc.) employ min-max affine quantization grid, which is suboptimal for preserving the quantized precision of the inverse-diagonal outliers, leading to high loss errors and model quality degradation. Our idea is to learn quantization grids that minimize the loss error $\epsilon$.

## 3.2 Loss-Error-Aware Network Quantization

Existing iterative loss-error-based quantization methods rely on min-max affine grids, which fail to account for outliers in the inverse Hessian diagonals. These outliers can cause significant degradation in model quality. To address this limitation, we propose loss-error-aware quantization grids that preserve the precision of weights corresponding to these outliers, thereby improving model quality. Our approach introduces techniques for learning loss-error-aware grids across various quantization formats, including non-uniform and affine. Additionally, to accelerate grid learning for large models, we developed fused GPU kernels that enable efficient and scalable quantization.

### 3.2.1 Non-Uniform Loss-Error-Aware Grid

For non-uniform quantization, we perform clustering on the model parameters, weighted by their corresponding exponentiated inverse Hessian diagonals, to derive a set of loss-error-aware grid

points. The proposed objective aims to shape the learned grid to minimize quantization error for weights corresponding to inverse-diagonal outliers, as these outliers can disproportionately affect model quality. Concretely, we determine the set of grid points $\mathcal{G}$ for $b$-bit quantization by optimizing the following objective:

$$\underset{\mathcal{G}:|\mathcal{G}|=2^b}{\arg\min} \sum_i (\mathbf{H}_{i,i}^{-1})^{-p} \left|\text{quant}_{nu}(w_i, \mathcal{G}) - w_i\right|^2 \tag{5}$$

Here, $p$ is a hyperparameter that balances the strength of precision preservation between inverse-diagonal outliers and non-outliers. Higher values of $p$ prioritize the precision preservation of outliers, while $p = 0$ treats all weights equally. In our experiments, we set $p = 4$ for all models. A sensitivity analysis for $p$ is provided in Section 4.3. To optimize this objective, we employ the k-means algorithm (Lloyd, 1982), incorporating careful centroid initialization as described below. Once the quantization grid $\mathcal{G}$ is established, the weights are iteratively quantized to the nearest grid points within $\mathcal{G}$.

**Grid Initialization** The quality of clustering relies heavily on initialization (Arthur et al., 2007), as Lloyd's Algorithm (Lloyd, 1982) converges to a locally optimal solution. This sensitivity is especially critical in low-bit-width settings (3-bit or 2-bit), where standard methods like random or k-means++ (Arthur et al., 2007) often undersample extreme values due to the distribution of weights, which are densely concentrated near the center and sparse at the extremes.

To address this, we propose **uniformly spaced grid initialization**, which evenly spaces initial grid points between the minimum and maximum weights to ensure both central regions and extremes are well represented. The initial grid points for clustering, $\mathcal{G}_{\text{init}}$, are defined as:

$$\mathcal{G}_{\text{init}} = \left\{ \min(\mathbf{w}) + \frac{\max(\mathbf{w}) - \min(\mathbf{w})}{2^b - 1} t \,\middle|\, t \in \{0, \ldots, 2^b - 1\} \right\} \tag{6}$$

This lightweight and robust initialization improves representation across the entire range of weights. We present an ablation study to confirm its effectiveness in Table 15 in the Appendix.

### 3.2.2 Loss-Error-Aware Affine Grid

The goal of learning an affine grid is to determine an optimal scaling factor $S$ and zero-point $Z$ that minimize the loss error. Unlike non-uniform grids, where clustering strategies can be applied, affine grids require the grid points to be uniformly spaced over an interval, making clustering-based approaches inapplicable. While gradient descent could be used to learn $S$ and $Z$, it is computationally intensive, memory-demanding, and susceptible to local minima.

To address this challenge, we adopt an enumerative search approach to learn the affine grid. Specifically, we enumerate candidate pairs of $S$ and $Z$ from a constrained search space $\mathbb{S}$ and select the pair that minimizes the following objective, which is similar to Equation 5:

$$\underset{(S,Z)\in\mathbb{S}}{\arg\min} \sum_i (\mathbf{H}_{i,i}^{-1})^{-p} \left|\text{quant}_{aff}(w_i, S, Z) - w_i\right|^2, \text{where}$$

$$\mathbb{S} = \left\{ \left( \underbrace{\frac{\left(\max(\mathbf{w}) - t_{\max}\frac{R}{T}\right) - \left(\min(\mathbf{w}) + t_{\min}\frac{R}{T}\right)}{2^b - 1}}_{\text{scaling factor } S}, \underbrace{-\left\lfloor \frac{\min(\mathbf{w}) + t_{\min}\frac{R}{T}}{S} \right\rceil}_{\text{zero-point } Z} \right) \middle| t_{\min}, t_{\max} \in \{0, \ldots, t\} \right\} \tag{7}$$

Here, $R = \max(\mathbf{w}) - \min(\mathbf{w})$ is the range of the weights, $T$ is the number of partitions within $R$, and $t \in \{1, \ldots, \frac{T}{2}\}$ is the number of partitions to enumerate over. By iteratively enumerating candidates of $S$ and $Z$ and evaluating their corresponding losses, we identify the optimal pair that minimizes the loss error. The parameter $T$ determines the granularity of the search; in our experiments, we set $T = 2048$. To prevent overfitting, $t$ controls the amount of shrinkage of the range, which we explain in Appendix B.

**Efficient Fused GPU Kernel for Grid Learning** The enumerative search for $S$ and $Z$ involves evaluating $t^2$ candidate pairs, which can be computationally expensive if performed sequentially. To accelerate this process, we design and implement a fused GPU kernel that leverages parallel

---

**Algorithm 1** LeanQuant for LLM quantization

---

**Input:** weight matrix $\mathbf{W} \in \mathbb{R}^{r \times c}$, input matrix $\mathbf{X}$, bit width $b$, block size $B$, dampening factor $df$, outlier preservation strength $p$
**Output:** Quantized matrix $\hat{\mathbf{W}}$

1: $\hat{\mathbf{W}} \leftarrow \mathbf{0}_{r \times c}$
2: $\mathbf{E} \leftarrow \mathbf{0}_{r \times B}$
3: $\mathbf{H} \leftarrow 2\mathbf{X}\mathbf{X}^{\top}$
4: $\mathbf{H}^{-1} \leftarrow \text{Cholesky}\left(\left[\mathbf{H} + df \cdot \text{avg}(\text{diag}(\mathbf{H})) \cdot \mathbf{I}\right]^{-1}\right)$        ▷ apply dampening, inversion, and Cholesky decomposition
5: **if** using non-uniform grid **then**
6:    $\mathcal{G}_k \leftarrow \underset{\mathcal{G}:|\mathcal{G}|=2^b}{\arg\min} \left(\text{diag}(\mathbf{H}^{-1})^{-p}\right)^{\top} \left|\text{quant}_{nu}(\mathbf{W}_{k,:}, \mathcal{G}) - \mathbf{W}_{k,:}\right|^2$ **forall** $k \in \{0, \ldots, r-1\}$        ▷ E.5
7: **else if** using affine grid **then**
8:    $S_k, Z_k \leftarrow \underset{(S,Z)\in\mathbb{S}}{\arg\min} \left(\text{diag}(\mathbf{H}^{-1})^{-p}\right)^{\top} \left|\text{quant}_{aff}(\mathbf{W}_{k,:}, S, Z) - \mathbf{W}_{k,:}\right|^2$ **forall** $k \in \{0, \ldots, r-1\}$        ▷ E.7
9: **end if**
10: **for** $i \leftarrow 0, B, 2B, \ldots$ **do**        ▷ apply block-wise quantization
11:    **for** $j \leftarrow i, \ldots, i+B-1$ **do**
12:       **if** using non-uniform grid **then**
13:          $\hat{\mathbf{W}}_{k,j} \leftarrow \text{quant}_{nu}(\mathbf{W}_{k,j}, \mathcal{G}_k)$ **forall** $k \in \{0, \ldots, r-1\}$        ▷ quantize to non-uniform grid
14:       **else if** using affine grid **then**
15:          $\hat{\mathbf{W}}_{k,j} \leftarrow \text{quant}_{aff}(\mathbf{W}_{k,j}, S_k, Z_k)$ **forall** $k \in \{0, \ldots, r-1\}$        ▷ quantize to affine grid
16:       **end if**
17:       $\mathbf{E}_{:,j-i} \leftarrow \dfrac{\mathbf{W}_{:,j} - \hat{\mathbf{W}}_{:,j}}{\mathbf{H}_{j,j}^{-1}}$
18:       $\mathbf{W}_{:,j:(i+B)} \leftarrow \mathbf{W}_{:,j:(i+B)} - \mathbf{E}_{:,j-i} \cdot \mathbf{H}_{j,j:(i+B)}^{-1}$
19:    **end for**
20:    $\mathbf{W}_{:,(i+B):} \leftarrow \mathbf{W}_{:,(i+B):} - \mathbf{E} \cdot \mathbf{H}_{i:(i+B),(i+B):}^{-1}$
21: **end for**
22: **return** $\hat{\mathbf{W}}$

---

processing. Each thread block is assigned a group of weights, and individual threads within the block evaluate all combinations of a specific $t_{\min}$ and all possible $t_{\max}$. The threads compute the loss for their assigned combinations, and the results are aggregated at the block level to determine the optimal $S$ and $Z$ for the weight group. This parallelized approach enables simultaneous computation of $S$ and $Z$ across all weight groups, achieving a speedup of over $50\times$ for the end-to-end quantization process. An analysis of the kernel's efficiency is presented in Section 4.3.

### 3.2.3 LeanQuant

Our proposed loss-error-aware quantization grid can be seamlessly integrated with any iterative loss-error-based quantization method to enhance the quality of quantized models. Figure 1 illustrates a comparison between the min-max affine quantization grid and loss-error-aware grids (both non-uniform and affine) applied to a layer of Llama-3-8B (Dubey et al., 2024). We introduce LeanQuant, which combines loss-error-aware grids with GPTQ (Frantar et al., 2022), and detail the method in Algorithm 1. Additionally, for quantizing million-parameter models more accurately, we propose LeanQuant-Exact, which integrates loss-error-aware grids with OBQ (Frantar & Alistarh, 2022), with details presented in Algorithm 2 in the Appendix. To specify the grid type used within Lean-Quant, we use subscripts such as LeanQuant$_{aff}$ for affine and LeanQuant$_{nu}$ for non-uniform grids.

## 4 Experiments

We conduct extensive experiments to validate LeanQuant's effectiveness and scalability in LLM quantization against competitive baselines. We first introduce the baselines, models, evaluation metrics, datasets, and hardware. Then, we present results, analyze efficiency and scalability, and conduct ablation studies to further validate our approach.

**Baselines** We compare LeanQuant$_{aff}$ against competitive affine quantization approaches AWQ (Lin et al., 2024), GPTQ (Frantar et al., 2022), and OmniQuant (Shao et al., 2024), and LeanQuant$_{nu}$ against the existing state-of-the-art non-uniform method SqueezeLLM (Kim et al., 2023). For the baselines, we use the quantized models provided by their official repository where possible, and quantize the unavailable models using their official codebase and recommended hyperparameters. More details on baseline reproduction and evaluation methods can be found in Section E of the Appendix. For LeanQuant models, we use a small calibration set of 128 sequences of 2048 tokens from the C4 dataset (Raffel et al., 2020) for computing the Hessian $\mathbf{H}$, and set $p = 4$.

Table 1: Zero-shot accuracy of quantized LLMs on benchmarks. The results of more models can be found in Table 10 of the Appendix. [†]2-bit quantization is unsupported by the SqueezeLLM codebase.

| | Method | Bits | ARC Easy | ARC Chg | LAMBADA Std | LAMBADA OpenAI | STEM | MMLU Human. | MMLU Social | MMLU Other | HellaS | PIQA | WinoG | Avg. |
|---|---|---|---|---|---|---|---|---|---|---|---|---|---|---|
| | **Llama-3-8B** | | | | | | | | | | | | | |
| | BF16 | 16 | 80.30 | 50.17 | 68.85 | 75.82 | 53.82 | 54.88 | 73.29 | 70.42 | 60.11 | 79.71 | 73.56 | 67.36 |
| Affine | GPTQ | 4.00 | 74.83 | 44.11 | 63.42 | 70.75 | 47.29 | 52.28 | 66.04 | 64.89 | 57.98 | 77.26 | 71.82 | 61.58 |
| Affine | OmniQuant | 4.00 | 76.89 | 47.35 | 61.05 | 69.16 | 49.38 | 49.05 | 66.62 | 64.40 | 58.25 | 78.84 | 71.98 | 63.00 |
| Affine | LeanQuant_aff | 4.00 | 76.60 | 46.93 | 66.89 | 74.07 | 51.89 | 52.96 | 70.04 | 68.43 | 58.47 | 77.91 | 72.77 | 65.18 |
| Affine | GPTQ | 3.00 | 50.84 | 24.32 | 24.16 | 38.89 | 26.23 | 29.16 | 34.38 | 30.00 | 45.07 | 64.64 | 60.69 | 37.75 |
| Affine | OmniQuant | 3.00 | 60.90 | 30.12 | 21.08 | 27.63 | 26.32 | 27.80 | 29.51 | 29.90 | 46.98 | 68.17 | 59.98 | 38.95 |
| Affine | LeanQuant_aff | 3.00 | 69.44 | 35.75 | 46.81 | 65.42 | 42.59 | 44.78 | 58.17 | 56.97 | 52.72 | 74.86 | 69.93 | 56.13 |
| Affine | GPTQ | 2.00 | 25.46 | 22.53 | 0.00 | 0.00 | 21.06 | 23.95 | 21.16 | 23.78 | 25.66 | 52.77 | 51.54 | 24.25 |
| Affine | OmniQuant | 2.00 | 26.81 | 21.67 | 0.00 | 0.00 | 21.34 | 24.21 | 21.71 | 23.98 | 25.90 | 53.75 | 47.43 | 24.26 |
| Affine | LeanQuant_aff | 2.00 | 35.06 | 18.26 | 11.33 | 14.71 | 21.31 | 24.17 | 21.71 | 24.01 | 31.43 | 59.30 | 51.85 | 28.47 |
| Non-uniform | SqueezeLLM | 4.05 | 79.59 | 49.32 | 66.18 | 73.24 | 51.13 | 53.32 | 70.78 | 68.59 | 59.10 | 79.33 | 73.80 | 65.85 |
| Non-uniform | LeanQuant_nu | 4.05 | 79.50 | 49.15 | 67.36 | 74.95 | 52.17 | 53.16 | 71.40 | 68.75 | 59.19 | 78.89 | 74.11 | 66.24 |
| Non-uniform | SqueezeLLM | 3.02 | 73.19 | 43.52 | 58.22 | 66.58 | 43.61 | 46.57 | 61.91 | 60.03 | 56.17 | 77.64 | 69.22 | 59.70 |
| Non-uniform | LeanQuant_nu | 3.02 | 77.74 | 47.01 | 63.32 | 72.17 | 48.84 | 49.05 | 65.45 | 62.79 | 56.42 | 78.24 | 71.67 | 62.97 |
| Non-uniform | SqueezeLLM[†] | 2.01 | | | | | | - N/A - | | | | | | |
| Non-uniform | LeanQuant_nu | 2.01 | 58.21 | 26.62 | 31.22 | 39.16 | 25.98 | 25.48 | 27.01 | 26.65 | 40.78 | 68.01 | 60.38 | 39.05 |
| | **Llama-2-7B** | | | | | | | | | | | | | |
| | FP16 | 16 | 76.26 | 43.43 | 68.33 | 73.88 | 34.38 | 39.79 | 47.32 | 47.12 | 57.10 | 78.07 | 68.98 | 57.70 |
| Affine | GPTQ | 4.00 | 74.16 | 40.78 | 65.38 | 71.94 | 32.67 | 36.92 | 42.61 | 42.61 | 55.99 | 77.48 | 68.32 | 53.47 |
| Affine | OmniQuant | 4.00 | 74.12 | 40.70 | 64.10 | 70.62 | 28.80 | 32.18 | 34.71 | 35.79 | 55.37 | 76.93 | 68.67 | 52.91 |
| Affine | LeanQuant_aff | 4.00 | 75.00 | 41.21 | 65.03 | 72.02 | 34.82 | 36.94 | 46.77 | 44.54 | 55.32 | 77.15 | 68.75 | 56.14 |
| Affine | GPTQ | 3.00 | 66.29 | 34.22 | 46.46 | 58.18 | 28.20 | 26.99 | 32.11 | 29.90 | 49.05 | 73.23 | 62.83 | 44.12 |
| Affine | OmniQuant | 3.00 | 70.12 | 37.29 | 53.27 | 66.66 | 29.05 | 31.05 | 30.61 | 30.38 | 52.58 | 74.05 | 66.46 | 49.23 |
| Affine | LeanQuant_aff | 3.00 | 69.28 | 37.12 | 59.77 | 67.73 | 30.32 | 30.22 | 35.26 | 33.34 | 50.59 | 74.81 | 66.14 | 50.42 |
| Affine | GPTQ | 2.00 | 25.97 | 21.67 | 0.00 | 0.00 | 21.31 | 23.25 | 21.11 | 23.01 | 25.76 | 51.74 | 48.78 | 23.66 |
| Affine | OmniQuant | 2.00 | 37.42 | 21.76 | 1.28 | 3.24 | 21.47 | 24.14 | 21.74 | 23.91 | 29.59 | 57.18 | 51.93 | 26.70 |
| Affine | LeanQuant_aff | 2.00 | 41.08 | 20.99 | 16.98 | 21.93 | 21.25 | 24.06 | 21.77 | 23.88 | 31.94 | 61.64 | 56.51 | 31.09 |
| Non-uniform | SqueezeLLM | 4.05 | 75.59 | 41.98 | 67.81 | 72.79 | 34.32 | 38.94 | 45.40 | 44.96 | 56.80 | 77.48 | 68.43 | 56.77 |
| Non-uniform | LeanQuant_nu | 4.05 | 75.97 | 42.66 | 68.14 | 74.25 | 34.35 | 39.06 | 46.05 | 46.51 | 56.03 | 77.86 | 69.38 | 57.30 |
| Non-uniform | SqueezeLLM | 3.02 | 73.06 | 40.27 | 61.96 | 70.11 | 33.75 | 35.22 | 43.35 | 43.16 | 54.15 | 76.50 | 67.88 | 54.49 |
| Non-uniform | LeanQuant_nu | 3.02 | 73.74 | 40.19 | 66.12 | 73.16 | 32.25 | 35.54 | 43.40 | 43.39 | 53.24 | 76.44 | 68.35 | 55.07 |
| Non-uniform | SqueezeLLM[†] | 2.01 | | | | | | - N/A - | | | | | | |
| Non-uniform | LeanQuant_nu | 2.01 | 51.81 | 23.98 | 28.68 | 38.21 | 22.26 | 23.89 | 22.49 | 24.01 | 35.88 | 66.38 | 58.17 | 35.98 |
| | **Mistral-7B** | | | | | | | | | | | | | |
| | BF16 | 16 | 80.77 | 50.09 | 69.38 | 75.63 | 50.46 | 53.48 | 69.35 | 68.01 | 61.26 | 80.58 | 73.88 | 66.62 |
| Affine | GPTQ | 4.00 | 79.00 | 46.25 | 66.99 | 73.67 | 46.24 | 50.82 | 66.20 | 64.66 | 59.36 | 79.65 | 72.93 | 62.68 |
| Affine | OmniQuant | 4.00 | 78.49 | 46.25 | 63.28 | 71.20 | 45.96 | 51.35 | 65.68 | 64.76 | 60.19 | 79.87 | 71.90 | 63.54 |
| Affine | LeanQuant_aff | 4.00 | 79.71 | 48.04 | 68.33 | 75.70 | 47.42 | 51.84 | 68.05 | 66.43 | 59.65 | 80.41 | 73.48 | 65.37 |
| Affine | GPTQ | 3.00 | 70.54 | 38.65 | 52.63 | 62.10 | 36.31 | 38.89 | 49.20 | 47.86 | 54.76 | 77.58 | 67.96 | 52.60 |
| Affine | OmniQuant | 3.00 | 70.54 | 35.07 | 35.49 | 46.54 | 33.71 | 32.88 | 40.23 | 37.85 | 52.35 | 75.19 | 63.93 | 47.62 |
| Affine | LeanQuant_aff | 3.00 | 77.65 | 44.71 | 60.51 | 71.94 | 43.99 | 46.14 | 60.97 | 59.35 | 55.61 | 78.51 | 71.59 | 61.00 |
| Affine | GPTQ | 2.00 | 26.73 | 22.27 | 0.00 | 0.00 | 23.31 | 24.46 | 23.86 | 23.42 | 25.35 | 51.52 | 49.72 | 24.39 |
| Affine | OmniQuant | 2.00 | 27.06 | 27.06 | 0.00 | 0.00 | 21.25 | 24.29 | 21.71 | 23.98 | 25.89 | 51.25 | 51.54 | 24.42 |
| Affine | LeanQuant_aff | 2.00 | 56.02 | 28.33 | 19.23 | 23.17 | 23.72 | 24.48 | 24.21 | 25.36 | 34.45 | 62.57 | 57.14 | 34.43 |
| Non-uniform | SqueezeLLM | 4.05 | 79.73 | 49.06 | 68.28 | 74.93 | 48.81 | 52.73 | 68.87 | 66.98 | 59.80 | 80.25 | 73.56 | 65.73 |
| Non-uniform | LeanQuant_nu | 4.05 | 79.80 | 48.89 | 69.03 | 76.03 | 48.84 | 52.86 | 68.87 | 66.69 | 60.19 | 80.14 | 74.59 | 65.99 |
| Non-uniform | SqueezeLLM | 3.02 | 77.54 | 45.93 | 64.06 | 71.43 | 43.96 | 47.93 | 62.69 | 59.16 | 58.76 | 79.43 | 71.98 | 62.08 |
| Non-uniform | LeanQuant_nu | 3.02 | 77.74 | 45.99 | 67.59 | 76.07 | 44.24 | 47.97 | 62.14 | 62.47 | 57.28 | 79.27 | 72.22 | 63.00 |
| Non-uniform | SqueezeLLM[†] | 2.01 | | | | | | - N/A - | | | | | | |
| Non-uniform | LeanQuant_nu | 2.01 | 63.47 | 30.55 | 41.01 | 54.61 | 31.34 | 29.97 | 32.14 | 33.96 | 42.29 | 71.38 | 64.01 | 44.97 |

**Models** We consider the following recent, popular LLMs for quantization: Llama 1/2/3 series models (Touvron et al., 2023a;b; Dubey et al., 2024), Mistral-7B-v0.1 (Jiang et al., 2023), Mistral-Large-Instruct-2407 (123B) (Mistral AI Team, 2024), and Llama-3.1-405B-Instruct (Dubey et al., 2024).

**Evaluation Metrics and Datasets** We evaluate quantized LLMs using the perplexity metric on the datasets WikiText2 (Merity et al., 2016) and C4 (Raffel et al., 2020), and zero-shot accuracy on the benchmarks ARC (Clark et al., 2018), LAMBADA (Paperno et al., 2016), MMLU (Hendrycks et al., 2020), HellaSwag (Zellers et al., 2019), PIQA (Bisk et al., 2020), and WinoGrande (Sakaguchi et al., 2021). We also quantize and evaluate the instruction-following Llama-3-8B-Instruct using OpenAI GPT-4o (2024-05-13) as a judge on the MT-Bench (Zheng et al., 2023), and the results are presented in Section G in the Appendix.

**Testbed Hardware** LeanQuant models are quantized using a machine quipped with an L40s-48GB GPU, an AMD EPYC 7R13 48-Core CPU, and 370GB of RAM. To fit Llama-3.1-405B-Instruct in RAM, which is around 800GB in size, we use a machine equipped with 2 Quadro RTX 8000 GPUs, an AMD EPYC 7742 64-Core CPU, and 1.48TB of RAM.

Table 2: Zero-shot accuracy of the quantized 123B Mistral-Large-Instruct-2407 model.

| Model | Method | Bits | Arc | | LAMBADA | | MMLU | | | | Avg. |
|---|---|---|---|---|---|---|---|---|---|---|---|
| | | | Easy | Chg. | Std. | OpenAI | STEM | Human. | Social | Other | |
| Mistral-Large-Instruct-2407 | GPTQ | 4.00 | 84.60 | 63.99 | 74.38 | 80.52 | 76.31 | 77.23 | 89.31 | 85.23 | 78.95 |
| | LeanQuant$_{aff}$ | 4.00 | 85.14 | 63.99 | 74.99 | 81.14 | **76.56** | 77.32 | 89.21 | **85.87** | 79.28 |
| | LeanQuant$_{nu}$ | 4.05 | **87.67** | **64.59** | **76.63** | **81.51** | 76.50 | **78.00** | **89.35** | 85.68 | **79.99** |

Table 3: Zero-shot accuracy of the quantized Llama-3.1-405B-Instruct model.

| Model | Method | Group Size | Bits | Arc | | LAMBADA | | MMLU | | | | Avg. |
|---|---|---|---|---|---|---|---|---|---|---|---|---|
| | | | | Easy | Chg. | Std. | OpenAI | STEM | Human. | Social | Other | |
| Llama-3.1-405B-Instruct | GPTQ | 128 | 4.25 | 88.21 | **65.10** | 73.41 | 76.96 | 82.34 | 82.64 | 90.45 | 87.51 | 80.83 |
| | LeanQuant$_{aff}$ | 128 | 4.25 | **88.26** | 64.76 | 73.32 | 77.08 | **82.68** | 83.21 | **90.58** | **87.71** | 80.95 |
| | LeanQuant$_{nu}$ | - | 4.05 | 88.22 | 63.65 | **74.56** | **78.52** | 82.52 | **83.40** | 90.51 | 87.64 | **81.13** |

## 4.1 MAIN RESULTS

**Accuracy and Perplexity** The zero-shot accuracy of quantized models on benchmarks are presented in Table 1, as well as in Table 10 in the Appendix, and the perplexity results are shown in Table 7 in the Appendix. At the same bit width, LeanQuant achieves significantly better (lower) perplexity than GPTQ and AWQ, and performs on par with OmniQuant and SqueezeLLM. However, perplexity may not be a representative metric for evaluating the accuracy of quantized models. In terms of zero-shot accuracy on various benchmarks, LeanQuant$_{aff}$ mostly outperforms GPTQ and OmniQuant, and LeanQuant$_{nu}$ similarly performs better than SqueezeLLM in most cases. We highlight that LeanQuant$_{aff}$ improves the average zero-shot accuracy on 11 tasks over OmniQuant by 17.18% for 3-bit Llama-3-8B, and by 13.38% for 3-bit Mistral-7B. Compared to GPTQ, LeanQuant$_{aff}$ improves the average zero-shot accuracy by 18.38% for 3-bit Llama-3-8B, and by 8.40% for 3-bit Mistral-7B.

**Effectiveness on Very Large LLMs** We quantize the 123B Mistral-Large-Instruct-2407 and the 405B Llama-3.1 model using LeanQuant$_{aff}$, LeanQuant$_{nu}$, and GPTQ, and present their zero-shot accuracy in Table 2 and 3, respectively. OmniQuant and SqueezeLLM fail to quantize to these models due to GPU out-of-memory errors. LeanQuant models mostly outperform GPTQ in zero-shot accuracy. For affine quantization, we employ row-wise quantization for Mistral-Large and group-wise quantization (with size 128) for Llama-3.1 405B. This showcases that our method is effective for both row-wise and group-wise quantization.

## 4.2 MEMORY AND TIME EFFICIENCY

We report the maximum GPU memory consumption of LeanQuant and the baselines during quantization on models of different sizes in Table 4. LeanQuant is significantly more memory efficient than OmniQuant and SqueezeLLM: it successfully scales to 123B Mistral-Large using a single 48GB GPU, and to 405B Llama-3.1 models using two 48GB GPUs, while OmniQuant fails to quantize Llama-3-70B and SqueezeLLM fails to quantize Llama-3-8B on a single 48GB GPU. The time cost of LeanQuant for different sized models are reported in Table 13 in the Appendix. LeanQuant can quantize 7B/8B models in less than an hour, the 123B model in 4.2 hours, and the 405B model in 20.7 hours.

## 4.3 ABLATION STUDY

**Q1: Does LeanQuant effectively reduce the loss error $\epsilon$ compared to other iterative loss-error-based methods?** Yes, LeanQuant effectively reduces loss errors $\epsilon$ compared to GPTQ, as shown in Figure 2, as well as in Figure 5 in the Appendix. The sum of loss errors are computed as Equation 3. Moreover, non-uniform LeanQuant generally achieves lower loss errors than affine LeanQuant, due to more degrees of freedom in the grid point placements, which also explains why LeanQuant$_{nu}$ achieves higher accuracy than LeanQuant$_{aff}$ on benchmarks in Table 1.

**Q2: Is LeanQuant sensitive to the hyperparameter $p$?** No, we found LeanQuant to be not very sensitive to $p$. A sensitivity analysis on the hyperparameter $p$ is given in Table 14 in the Appendix. LeanQuant works well with $p$ values of 3 or 4.

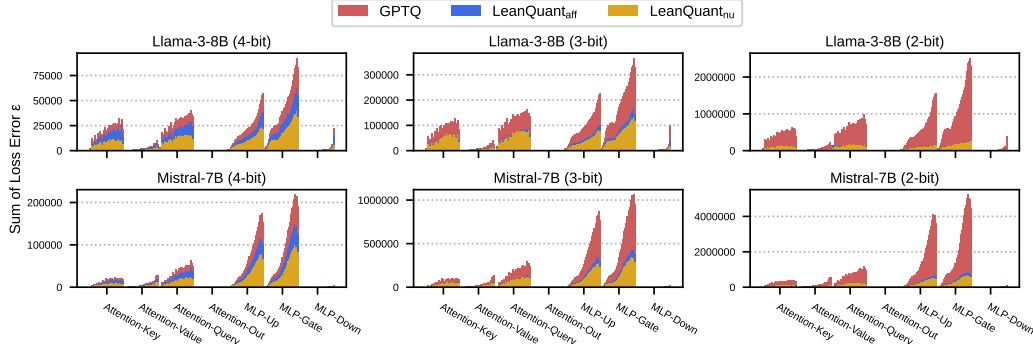

Figure 2: Comparison of loss errors $\epsilon$, summed over each layer, for GPTQ and LeanQuant (affine and non-uniform) during iterative quantization.

**Q3: Is uniformly spaced grid initialization beneficial for model quality?** Yes, uniformly spaced grid initialization consistently outperforms k-means++ (Arthur et al., 2007) initialization on different models in 3-bit and 2-bit regions, as shown in Table 15 in the Appendix.

**Q4: Does the fused GPU kernel for LeanQuant$_{aff}$ accelerate quantization?** Yes, our fused kernel for learning affine grids accelerate the end-to-end quantization process by more than $50\times$, as shown in Table 5, which enables LeanQuant to be scaled to very large models.

Table 4: Peak GPU memory consumption of different algorithms during 4-bit quantization. "OOM" indicates out of memory on a single 48GB GPU, except for Llama-3.1-405B where we use 2 48GB GPUs.

| Model | OmniQuant | SqueezeLLM | GPTQ | LeanQuant |
|---|---|---|---|---|
| Llama-3-8B | 25.3 GB | OOM | 7.9 GB | 7.9 GB |
| Llama-3-70B | OOM | OOM | 17.1 GB | 17.2 GB |
| Mistral-Large (123B) | OOM | OOM | 32.8 GB | 33.0 GB |
| Llama-3.1-405B | OOM | OOM | OOM | 65.4 GB |

Table 5: Comparison of total time needed for quantizing Llama-3-8B with and without our fused kernel for loss-error-aware affine grid learning.

| Fused Kernel | Group Size | Bits | Quant. Time |
|---|---|---|---|
| ✗ | - | 4.00 | 15.1 hrs |
| ✓ | - | 4.00 | 0.27 hrs |
| ✗ | 128 | 4.25 | >100 hrs |
| ✓ | 128 | 4.25 | 0.40 hrs |

## 5 RELATED WORKS

**Iterative Loss-error-based Compression** Optimal Brain Damage (LeCun et al., 1989) introduced a saliency-score-based iterative pruning algorithm for neural networks, and Optimal Brain Surgeon (Hassibi & Stork, 1992; Hassibi et al., 1993) extended it to apply a weight update to compensate for the error introduced in each iteration. These methods inspired a number of works on model pruning (Guo et al., 2016; Singh & Alistarh, 2020; Yu et al., 2022) and weight quantization (Li et al., 2021; Frantar & Alistarh, 2022; Frantar et al., 2022).

**Efficient LLM Inference** LLM inference is computationally and memory demanding, and existing works explore improving inference efficiency through quantization (Dettmers et al., 2022b; Lin et al., 2024; Frantar et al., 2022; Chee et al., 2024; Kim et al., 2023; Shao et al., 2024; Egiazarian et al., 2024; Tseng et al., 2024), pruning (Frantar & Alistarh, 2023; Ashkboos et al., 2024), weight-activation quantization (Xiao et al., 2023), offloading Sheng et al. (2023), etc. A survey of more relevant literature can be found in Appendix M.

## 6 CONCLUSION

In this work, we propose LeanQuant, an accurate, versatile, and scalable quantization method for LLMs. Motivated by the finding that the min-max affine grid causes large errors in the network's task loss in iterative loss-error-based methods, we propose to learn loss-error-aware grids to enable more accurate quantized models, and design fused kernels for efficient and scalable quantization. Our method generalizes to multiple quantization formats to enable greater accessibility. Extensive empirical evaluations reveal that our quantized models compares favorably against competitive baselines in accuracy, and can scale to Llama-3.1 405B, one of the largest open-source LLM to date.

ACKNOWLEDGEMENTS

This work was supported by National Science Foundation SHF-2211815 and Ken Kennedy Institute Cluster Grants.

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

APPENDIX

## A  EXPLANATIONS ON QUANTIZATION GRID

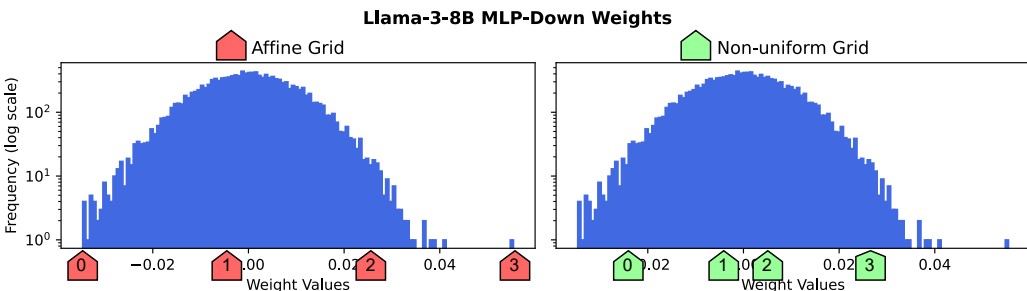

Figure 3: Comparison of affine (left) and non-uniform (right) 2-bit quantization grids applied to the weights in the first MLP-down layer of Llama-3-8B. The affine grid uses evenly spaced quantization grid points between the minimum and maximum weights. In contrast, the non-uniform grid allows grid points to be placed flexibly, as their positions are stored in a look-up table. This enables finer quantization in dense regions and coarser quantization in sparse regions, better aligning with the weight distribution and reducing quantization error.

In the context of quantization, a grid is a predefined set of values representing the possible quantized outputs for full-precision parameters. During quantization, each full-precision parameter is mapped to its nearest grid point on the quantization grid. For example, in a 2-bit quantization scheme with grid points $\{-1.0, -0.33, 0.33, 1.0\}$, a floating-point weight of 0.25 would be assigned to 0.33, the closest grid point.

**Affine Quantization Grid**  An affine quantization grid distributes points uniformly across the range of the weights being quantized. The dynamic range of the weights, defined as $[W_{\min}, W_{\max}]$, determines the spacing of the grid points. For example, if $[W_{\min}, W_{\max}] = [-1.0, 1.0]$ in a 2-bit quantization setting, the grid points would be evenly spaced at $-1.0, -0.33, 0.33, 1.0$. This uniform distribution is computationally simple and widely used in practice, but it may lead to suboptimal precision when the weight distribution is non-uniform, as many grid points may be underutilized.

**Non-uniform Quantization Grid**  Non-uniform grids allocate grid points more flexibly, allowing denser spacing in high-probability regions of the weight distribution and sparser spacing in low-probability regions. This approach minimizes quantization error by adapting the grid to the data distribution. Non-uniform grids typically store the grid points in a look-up table, enabling flexible placement that better represents the original data. Figure 3 illustrates an example of affine grid and non-uniform grid applied to the weights of Llama-3-8B.

**Grouped Quantization**  The quantization grid for a set of weights is determined by the range $[W_{\min}, W_{\max}]$ within the group. Smaller group sizes allow for a narrower dynamic range, leading to finer granularity in the quantization grid and higher precision. Grouping contiguous weights into blocks is a common practice in quantization literature (Lin et al., 2024; Frantar et al., 2022) and ensures a balance between memory efficiency and precision.

## B  CONTROLLING RANGE SHRINKAGE

In Equation 7, we enumerate candidate pairs $(S, Z)$—scaling factors and zero-points—to determine the optimal loss-error-aware affine quantization grid. This process involves iteratively refining $S$ and $Z$ by reducing the maximum value $\max(\mathbf{w})$ and increasing the minimum value $\min(\mathbf{w})$. However, excessive shrinking of the range may result in poor representation of extreme values, leading to model quality degradation.

---

**Algorithm 2** LeanQuant-Exact for Millon-parameter Networks

---

    **Input:** a row $\mathbf{w} \in \mathbb{R}^c$ in the weight matrix, sample input matrix $\mathbf{X}$, bit width $b$, hyperparameter $p$
    **Output:** Quantized row $\hat{\mathbf{w}}$
1:  $\hat{\mathbf{w}} \leftarrow \mathbf{0}_c$
2:  $\mathbf{H}^{-1} \leftarrow (2\mathbf{X}\mathbf{X}^\top)^{-1}$
3:  **if** using non-uniform grid **then**
4:    $\mathcal{G} \leftarrow \underset{\mathcal{G}:|\mathcal{G}|=2^b}{\arg\min} \left(\mathrm{diag}(\mathbf{H}^{-1})^{-p}\right)^\top \left|\mathrm{quant}_{nu}(\mathbf{w}, \mathcal{G}) - \mathbf{w}\right|^2$                     $\triangleright$ E. 5
5:  **else if** using affine grid **then**
6:    $S, Z \leftarrow \underset{(S,Z)\in\mathbb{S}}{\arg\min} \left(\mathrm{diag}(\mathbf{H}^{-1})^{-p}\right)^\top \left|\mathrm{quant}_{aff}(\mathbf{w}, S, Z) - \mathbf{w}\right|^2$           $\triangleright$ E. 7
7:  **end if**
8:  **for** $j \leftarrow 1, \ldots, c$ **do**
9:    **if** using non-uniform grid **then**
10:      $i \leftarrow \arg\min_i \frac{(\mathrm{quant}_{nu}(w_i, \mathcal{G}) - w_i)^2}{2\mathbf{H}_{i,i}^{-1}}$
11:      $\hat{w}_i \leftarrow \mathrm{quant}_{nu}(w_i, \mathcal{G})$
12:    **else if** using affine grid **then**
13:      $i \leftarrow \arg\min_i \frac{(\mathrm{quant}_{aff}(w_i, S, Z) - w_i)^2}{2\mathbf{H}_{i,i}^{-1}}$
14:      $\hat{w}_i \leftarrow \mathrm{quant}_{aff}(w_i, S, Z)$
15:    **end if**
16:    $\mathbf{w} \leftarrow \mathbf{w} - \frac{\mathbf{H}_{:,i}^{-1}}{\mathbf{H}_{i,i}^{-1}}\left(w_i - \hat{w}_i\right)$
17:    $\mathbf{H}^{-1} \leftarrow \mathbf{H}^{-1} - \frac{\mathbf{H}_{:,i}^{-1}\mathbf{H}_{i,:}^{-1}}{\mathbf{H}_{i,i}^{-1}}$
18: **end for**
19: **return** $\hat{\mathbf{w}}$

---

To control the extent of range reduction, we introduce the parameter $t$, which determines the degree of shrinkage. Lower bit widths require more aggressive shrinking due to the limited number of grid points. We set $t$ for $b$-bit quantization as follows:

$$t = \begin{cases} 0.2T & \text{if } b = 4, \\ 0.3T & \text{if } b = 3, \\ 0.4T & \text{if } b = 2. \end{cases} \tag{8}$$

## C   LEANQUANT-EXACT

The pseudocode of LeanQuant-Exact for accurately quantizing million-parameter networks is presented in Algorithm 2.

### C.1   BERT EXPERIMENTS WITH LEANQUANT-EXACT

| Method | Bits | BERT-3 | BERT |
|---|---|---|---|
| FP32 | 32 | 84.66 | 88.53 |
| OBQ | 4.03 | 84.40 | 87.96 |
| LeanQuant$_{nu}$-Exact | 4.13 | **84.58** | **88.49** |
| OBQ | 3.03 | 83.47 | 84.72 |
| LeanQuant$_{nu}$-Exact | 3.06 | **84.20** | **86.21** |

Table 6: F1 scores on SQuAD of BERT models quantized using OBQ and LeanQuant$_{nu}$-Exact. LeanQuant$_{nu}$-Exact outperforms OBQ in maintaining model quality.

We compare the performance of BERT models (Devlin et al., 2018), quantized with OBQ (Frantar & Alistarh, 2022) and LeanQuant$_{nu}$-Exact, on the SQuAD dataset (Rajpurkar et al., 2016). We quantize the 12-layer BERT-base (Devlin et al., 2018) and the 3-layer BERT-3 variant from Kurtic

Table 7: Perplexity evaluations of Llama models under different quantization methods and bit widths. The results of GPTQ, AWQ, OmniQuant are from Shao et al. (2024), and the results of SqueezeLLM are from Kim et al. (2023). [†] The official SqueezeLLM code does not support 2-bit quantization, and we report the available results from Kim et al. (2023).

| Grid | Method | Bits | WikiText-2 | | | | | C4 | | | | | Avg. |
|---|---|---|---|---|---|---|---|---|---|---|---|---|---|
| | | | 1-7B | 1-13B | 2-7B | 2-13B | 2-70B | 1-7B | 1-13B | 2-7B | 2-13B | 2-70B | |
| | FP16 | 16 | 5.58 | 5.09 | 5.47 | 4.88 | 3.31 | 7.08 | 6.61 | 6.97 | 6.46 | 5.52 | 5.697 |
| Affine | GPTQ | 4.00 | 6.13 | 5.40 | 5.83 | 5.13 | 3.58 | 7.43 | 6.84 | 7.37 | 6.70 | 5.67 | 6.008 |
| | AWQ | 4.00 | 6.08 | 5.34 | 6.15 | 5.12 | - | 7.52 | 6.86 | 7.68 | 6.74 | - | - |
| | OmniQuant | 4.00 | 5.86 | 5.21 | 5.74 | 5.02 | 3.47 | 7.34 | 6.76 | 7.35 | 6.65 | 5.65 | 5.905 |
| | LeanQuant$_{aff}$ | 4.00 | 5.92 | 5.25 | 5.73 | 5.08 | 3.49 | 7.30 | 6.76 | 7.25 | 6.63 | 5.63 | **5.904** |
| Non-uniform | SqueezeLLM | 4.04-4.05 | 5.79 | 5.18 | 5.62 | 4.99 | 3.41 | 7.21 | 6.71 | 7.12 | 6.57 | 5.58 | **5.818** |
| | LeanQuant$_{nu}$ | 4.04-4.05 | 5.81 | 5.19 | 5.64 | 4.99 | 3.42 | 7.21 | 6.70 | 7.13 | 6.57 | 5.58 | 5.824 |
| Affine | GPTQ | 3.00 | 8.06 | 6.76 | 8.37 | 6.44 | 4.82 | 9.49 | 8.16 | 9.81 | 8.02 | 6.57 | 7.650 |
| | AWQ | 3.00 | 11.88 | 7.45 | 24.00 | 10.45 | - | 13.26 | 9.13 | 23.85 | 13.07 | - | - |
| | OmniQuant | 3.00 | 6.49 | 5.68 | 6.58 | 5.58 | 3.92 | 8.19 | 7.32 | 8.65 | 7.44 | 6.06 | 6.591 |
| | LeanQuant$_{aff}$ | 3.00 | 6.62 | 5.76 | 6.61 | 5.66 | 3.91 | 7.98 | 7.19 | 8.27 | 7.23 | 5.90 | **6.513** |
| Non-uniform | SqueezeLLM | 3.02 | 6.32 | 5.60 | 6.18 | 5.36 | 3.77 | 7.75 | 7.08 | 7.72 | 6.97 | 5.83 | **6.258** |
| | LeanQuant$_{nu}$ | 3.02 | 6.34 | 5.60 | 6.19 | 5.40 | 3.80 | 7.74 | 7.05 | 7.73 | 6.98 | 5.83 | 6.266 |
| Affine | GPTQ | 2.00 | 1.1E5 | 6.8E4 | 3.8E4 | 5.6E4 | 2.0E4 | 689.13 | 2.5E3 | NaN | 323.12 | 48.82 | NaN |
| | OmniQuant | 2.00 | 15.47 | 13.21 | 37.37 | 17.21 | 7.81 | 24.89 | 18.31 | 90.64 | 26.76 | 12.28 | 26.395 |
| | LeanQuant$_{aff}$ | 2.00 | 18.53 | 14.42 | 25.69 | 24.43 | 7.92 | 19.99 | 16.53 | 27.11 | 20.92 | 10.84 | **18.638** |
| Non-uniform | SqueezeLLM† | 2.01 | | - N/A - | | 61.25 | 10.86 | | | - N/A - | | | N/A |
| | LeanQuant$_{nu}$ | 2.01 | 15.65 | 9.64 | 15.51 | 10.06 | 6.35 | 17.62 | 10.93 | 17.07 | 11.83 | 7.96 | **12.262** |

et al. (2022) to 3 and 4 bits. OBQ and LeanQuant-Exact are calibrated using 1024 samples from the training set, and the F1 score is reported on the test set.

## D    ROBUSTNESS OF LEANQUANT GRIDS DURING QUANTIZATION

LeanQuant prevents drastic increase to the task loss by learning the quantization grid for better preservation of the precision of outlier inverse diagonals. However, since the not-yet-quantized weights will shift during the iterative quantization process and the quantization grid is fixed beforehand, one potential problem arises: the quantization grid may no longer be well-aligned with the outliers after certain iterations. Fortunately, this is not a problem in practice. The loss-error-awareness property of LeanQuant grids prevents high-norm weight perturbations $\delta_i$ (Equation 3) from ocurring, hence the weights do not shift by much during the iterations. Furthermore, no new inverse-diagonal outliers will arise during the iterative quantization process. In OBQ, the inverse Hessian is updated after each iteration as follows,

$$\mathbf{H}_{-i,-i}^{-1} = \left( \mathbf{H}^{-1} - \frac{\mathbf{H}_{:,i}^{-1}\mathbf{H}_{i,:}^{-1}}{\mathbf{H}_{i,i}^{-1}} \right)_{-i,-i} \tag{9}$$

where $\mathbf{H}_{-i,-i}^{-1}$ is the inverse Hessian with its $i$-th row and column removed. The remaining inverse diagonals only decrease in magnitude towards zero after each column and row removal.

## E    EXPERIMENT DETAILS

**Baseline Reproduction**   We use the quantized models provided by the official repository where possible. We obtained quantized LLaMA-7B, LLaMA-13B, Llama-2-7B, Llama-2-13B from the OmniQuant repository, and LLaMA-7B, LLaMA-13B, Llama-2-7B, Llama-2-13B, Mistral-7B from the SqueezeLLM repository. We obtained the community-driven GPTQ-quantized version of Llama-3.1-405B-Instruct from HuggingFace [1]. The other quantized models are reproduced using the official codebases and recommended hyperparameters. For OmniQuant, we set the training

---
[1] `https://huggingface.co/hugging-quants/Meta-Llama-3.1-405B-Instruct-GPTQ-INT4`

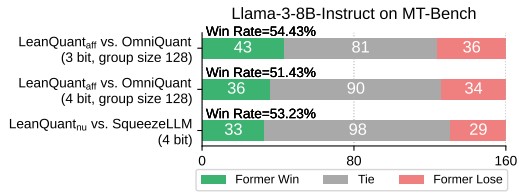

Figure 4: Evaluation of quantized Llama-3-8B-Instruct on MT-Bench using OpenAI GPT-4o as a judge. The win rates reported exclude ties.

Table 8: Zero-shot accuracy comparison between LeanQuant$_{aff}$ and rotation-based quantization method QuaRot.

| | Method | Bits | ARC | | LAMBADA | | MMLU | | | | Avg. |
| | | | Easy | Chg | Std | OpenAI | STEM | Human. | Social | Other | |
|---|---|---|---|---|---|---|---|---|---|---|---|
| Llama-3-8B | QuaRot-RTN | 4.00 | 65.92 | 46.93 | 65.75 | 71.92 | 49.67 | 51.24 | 65.75 | 64.24 | 60.18 |
| | QuaRot-GPTQ | 4.00 | **76.81** | 50.43 | 67.53 | 74.19 | **51.60** | 53.28 | 70.26 | 68.23 | 64.04 |
| | LeanQuant$_{aff}$ | 4.00 | 76.30 | **50.51** | **68.27** | **75.90** | 51.51 | **53.43** | **70.69** | **68.43** | **64.38** |
| Llama-2-13B | QuaRot-RTN | 4.00 | 74.54 | 47.44 | 66.19 | 74.17 | 39.74 | 41.74 | 53.04 | 51.95 | 56.10 |
| | QuaRot-GPTQ | 4.00 | 76.35 | **49.23** | 69.82 | **76.48** | 41.29 | 47.01 | 58.99 | 57.84 | 59.63 |
| | LeanQuant$_{aff}$ | 4.00 | **76.43** | 49.06 | **69.90** | **76.48** | **42.53** | 47.86 | **60.03** | **58.93** | **60.15** |

epochs to 20, enable Learnable Weight Clipping (LWC), set an LWC learning rate of 1e-2. For SqueezeLLM, there is no tunable parameters. For GPTQ, we turn on activation ordering (quantizing columns in order of decreasing activation size) for more accurate model.

**Perplexity Evaluations**   We follow the perplexity evaluation procedure described by (Frantar et al., 2022): sequences from the test set of the WikiText2 and C4 datasets (Merity et al., 2016; Raffel et al., 2020) are concatenated into 128 sequences of length 2048 tokens for perplexity testing.

**Accuracy Evaluations**   We use lm-evaluation-harness (Gao et al., 2023) for evaluating zero-shot accuracy on tasks. The task names we evaluate are `lambada`, `ai2_arc`, `winogrande`, `piqa`, `hellaswag`, `mmlu`.

## F   PERPLEXITY EVALUATIONS

The perplexity evaluation results on WikiText2 (Merity et al., 2016) and C4 (Raffel et al., 2020) for quantized models are presented in Table 7.

## G   LLM-AS-A-JUDGE

**LLM as a Judge** The evaluation results on MT-Bench using GPT-4o (2024-05-13) as a judge are presented in Figure 4. We pitch 3-bit and 4-bit, with group size of 128, LeanQuant$_{aff}$ against OmniQuant, and 4-bit LeanQuant$_{nu}$ against SqueezeLLM. LeanQuant achieves higher win rate than the baselines.

## H   MORE ACCURACY RESULTS

The zero-shot accuracy results on benchmarks for quantized LLaMA-7B, LLaMA-13B, Llama-2-7B (Touvron et al., 2023a;b) are presented in Table 10. We also compare affine LeanQuant with the rotation-based quantization algorithm QuaRot (Ashkboos et al., 2025), with results presented in Table 8. Furthermore, we compare affine, group-wise quantization using LeanQuant$_{aff}$, OmniQuant, and AWQ in Table 9.

Table 9: Zero-shot accuracy of affine, group-wise quantized models using LeanQuant$_{aff}$, Omni-Quant, and AWQ.

| Method | Group Size | Bits | ARC | | LAMBADA | | MMLU | | | | Avg. |
|--------|-----------|------|-----|-----|-----|--------|------|--------|--------|-------|------|
| | | | Easy | Chg | Std | OpenAI | STEM | Human. | Social | Other | |
| **Llama-2-7B** | | | | | | | | | | | |
| OmniQuant | 128 | 4.25 | 75.21 | **43.69** | 66.95 | 72.91 | 35.11 | 37.79 | 46.77 | 46.86 | 53.16 |
| AWQ | 128 | 4.25 | 75.17 | 43.26 | 67.40 | 72.70 | 34.89 | 37.79 | 46.34 | 45.93 | 52.94 |
| LeanQuant$_{aff}$ | 128 | 4.25 | **76.26** | 43.00 | **67.75** | **74.44** | **35.65** | **39.02** | **47.94** | **49.79** | **54.23** |
| **Llama-2-13B** | | | | | | | | | | | |
| OmniQuant | 128 | 4.25 | 78.24 | **47.61** | 69.75 | 75.99 | **42.06** | 47.74 | **59.83** | 58.61 | 59.98 |
| AWQ | 128 | 4.25 | 78.91 | 46.50 | 70.17 | 76.19 | 41.39 | 46.70 | 59.38 | 55.94 | 59.40 |
| LeanQuant$_{aff}$ | 128 | 4.25 | **79.00** | 47.18 | **70.75** | **77.95** | 41.96 | **47.86** | 59.54 | **58.97** | **60.40** |
| **Llama-3.1-70B-Instruct** | | | | | | | | | | | |
| AWQ | 128 | 4.25 | **86.62** | **62.12** | 72.09 | 75.70 | 74.88 | 80.60 | 87.07 | **83.71** | 77.85 |
| LeanQuant$_{aff}$ | 128 | 4.25 | 86.49 | 61.69 | **72.70** | **76.07** | **75.61** | **80.79** | **87.55** | **83.71** | **78.08** |

## I QUANTIZATION COST AND OVERHEAD

The time cost of LeanQuant for different models and configurations are presented in Table 13. A comparison of GPU memory consumption for different quantization algorithms on different-sized LLMs is presented in Table 11.

## J INFERENCE EFFICIENCY OF QUANTIZED MODELS

Table 12 presents the inference efficiency of 4-bit quantized Llama-3-8B during the decoding and prefill phases. For non-uniform LeanQuant models, we have developed a dedicated CUDA kernel for efficient inference, and we compare its efficiency against the SqueezeLLM kernel in Table 12. For affine LeanQuant models, we leverage the `exllamav2` kernels (Turboderp-org, 2024).

The inference efficiency in Table 12 is evaluated on an NVIDIA A100-40GB GPU. For decoding, we report *tokens per second per batch* while generating 4096 tokens. For the prefill phase, we measure *time to first token* using a 4096-token prompt.

## K ABLATION STUDY

**Sensitivity to Hyperparameter** $p$ Ablative experiments on the effects of the hyperparameter $p$ on the quality of LeanQuant models are presented in Table 14. In the case of $p = 0$, the inverse Hessian diagonals are ignored as the weights for clustering, and the centroids are learned based on the density of weights. It is worth noting that $p = 0$ results in sub-optimal model quality compared to higher values of $p$, which means that the loss-error-awareness property of the quantization grid is critical for maintaining model quality.

**Grid Point Initialization** Ablative experiments comparing k-means++ initialization with our proposed uniformly spaced grid initialization are presented in Table 15.

## L LOSS ERROR COMPARISON

A comparison of the sum of loss errors $\epsilon$ between GPTQ and LeanQuant (affine and non-uniform) is presented in Figure 5.

## M MORE RELATED WORKS

**Quantization for Large Language Models** Quantization reduces the precision of LLM parameters to achieve model compression and enable more memory-efficient inference. Calibration-free

Table 10: Zero-shot accuracy of more quantized LLMs on benchmarks.

| | Method | Bits | ARC | | LAMBADA | | MMLU | | | | HellaS | PIQA | WinoG | Avg. |
|---|---|---|---|---|---|---|---|---|---|---|---|---|---|---|
| | | | Easy | Chg | Std | OpenAI | STEM | Human. | Social | Other | | | | |
| | **LLaMA-7B** | | | | | | | | | | | | | |
| | FP16 | 16 | 75.29 | 41.81 | 67.77 | 73.49 | 28.20 | 32.03 | 31.65 | 36.53 | 56.92 | 78.67 | 70.09 | 53.86 |
| Affine | GPTQ | 4.00 | 73.61 | 39.51 | **65.61** | **71.98** | 24.29 | 28.12 | 25.80 | 31.16 | 55.61 | 77.80 | **70.40** | 49.03 |
| | OmniQuant | 4.00 | **74.49** | 39.68 | 65.38 | 71.96 | 30.32 | 30.92 | **33.38** | 35.79 | 55.82 | **78.45** | 68.67 | 53.17 |
| | LeanQuant$_{aff}$ | 4.00 | 74.03 | **41.64** | 63.75 | 70.56 | **31.27** | **33.13** | 32.66 | **37.79** | **56.31** | 78.40 | 69.30 | **53.53** |
| | GPTQ | 3.00 | 66.67 | 35.84 | 49.89 | 58.37 | 25.94 | 27.46 | 23.46 | 24.98 | 51.76 | 75.35 | 64.72 | 43.78 |
| | OmniQuant | 3.00 | 72.22 | 38.48 | 59.67 | 69.20 | 27.18 | 27.65 | 25.64 | 29.61 | 52.99 | 76.55 | 67.17 | 49.67 |
| | LeanQuant$_{aff}$ | 3.00 | **73.70** | **40.78** | **65.28** | **72.66** | **27.85** | **31.37** | **30.61** | **35.31** | **56.23** | **78.45** | **70.09** | **52.94** |
| | GPTQ | 2.00 | 26.35 | 22.01 | 0.00 | 0.00 | 23.69 | 25.08 | 24.28 | 24.01 | 25.69 | 53.70 | 50.99 | 24.95 |
| | OmniQuant | 2.00 | 51.05 | 22.70 | 11.80 | 23.13 | **26.51** | **26.04** | **24.37** | 23.69 | 34.71 | 64.36 | 54.30 | 32.97 |
| | LeanQuant$_{aff}$ | 2.00 | **55.98** | **27.22** | **38.56** | **47.45** | 24.45 | 25.31 | 22.85 | **25.46** | **37.19** | **68.12** | **60.77** | **39.40** |
| Non-uniform | SqueezeLLM | 4.05 | 74.92 | 40.61 | 66.87 | 71.99 | 29.38 | 29.86 | 29.38 | 34.86 | **56.55** | **78.29** | 69.38 | 52.89 |
| | LeanQuant | 4.05 | **75.17** | **41.55** | **69.01** | **74.33** | 27.37 | **30.50** | **29.41** | **35.44** | 56.09 | 77.91 | **70.17** | **53.36** |
| | SqueezeLLM | 3.02 | 71.46 | **37.88** | 61.71 | 71.05 | 23.66 | 26.99 | 26.16 | 30.54 | **54.89** | **77.86** | 68.59 | 50.07 |
| | LeanQuant | 3.02 | **71.76** | 36.35 | **63.77** | **71.71** | 26.64 | **29.05** | **29.80** | **32.70** | 53.80 | 77.09 | **69.53** | **51.11** |
| | SqueezeLLM | 2.01 | | | | | - N/A - | | | | | | | |
| | LeanQuant$_{nu}$ | 2.01 | 50.38 | 24.40 | 31.67 | 41.30 | 21.79 | 24.19 | 21.74 | 24.36 | 37.49 | 65.67 | 58.33 | 36.48 |
| | **LLaMA-13B** | | | | | | | | | | | | | |
| | FP16 | 16 | 77.40 | 46.42 | 71.12 | 76.19 | 36.41 | 41.55 | 48.49 | 48.54 | 59.92 | 79.16 | 72.69 | 59.81 |
| Affine | GPTQ | 4.00 | 77.06 | 45.56 | 69.12 | 75.28 | 34.44 | 39.15 | 45.95 | 46.73 | 58.99 | 78.56 | 72.53 | 56.63 |
| | OmniQuant | 4.00 | 75.97 | 45.22 | 68.25 | 75.59 | 35.30 | **40.21** | **48.20** | **47.25** | **59.11** | **78.94** | 72.61 | 58.79 |
| | LeanQuant$_{aff}$ | 4.00 | **76.39** | **46.42** | **70.48** | **76.27** | **35.52** | 39.45 | 46.18 | 47.22 | 58.82 | **78.94** | **72.30** | **58.91** |
| | GPTQ | 3.00 | 70.92 | 39.93 | 57.29 | 64.82 | 29.37 | 31.94 | 33.18 | 35.34 | 54.05 | 76.99 | 68.43 | 49.13 |
| | OmniQuant | 3.00 | 75.42 | 42.83 | 60.80 | 71.34 | 29.56 | 34.24 | 36.24 | 41.17 | **57.27** | **77.97** | 69.61 | 54.22 |
| | LeanQuant$_{aff}$ | 3.00 | **75.84** | **43.34** | **67.49** | **74.85** | **33.37** | **36.56** | **41.92** | **44.74** | 56.75 | **77.97** | **70.48** | **56.66** |
| | GPTQ | 2.00 | 27.10 | 21.93 | 0.02 | 0.00 | 23.37 | 25.50 | 23.56 | 24.49 | 25.76 | 53.16 | 49.72 | 24.75 |
| | OmniQuant | 2.00 | 59.51 | **29.52** | 17.85 | 23.35 | 22.52 | 24.12 | 22.81 | 24.46 | **42.01** | 67.25 | 56.12 | 35.41 |
| | LeanQuant$_{aff}$ | 2.00 | **61.45** | 29.27 | **44.63** | **50.82** | **28.73** | **26.01** | **27.20** | **27.26** | 39.08 | **71.27** | **65.82** | **42.87** |
| Non-uniform | SqueezeLLM | 4.04 | **76.56** | **46.25** | 69.53 | 75.22 | 32.98 | 37.19 | 42.18 | 44.00 | 59.29 | 78.62 | 71.82 | 57.60 |
| | LeanQuant$_{nu}$ | 4.04 | 76.39 | 45.05 | **71.55** | **76.48** | **34.76** | **38.77** | **46.12** | **47.18** | 59.29 | **78.78** | **73.24** | **58.87** |
| | SqueezeLLM | 3.02 | **75.46** | **43.77** | 65.07 | 72.75 | 30.45 | 34.24 | 37.18 | 40.46 | 57.32 | **78.29** | **71.35** | 55.12 |
| | LeanQuant$_{nu}$ | 3.02 | 75.17 | 43.00 | **70.41** | **77.29** | **33.81** | **38.32** | **43.16** | **45.06** | **57.35** | **78.29** | **71.35** | **57.56** |
| | SqueezeLLM | 2.01 | | | | | - N/A - | | | | | | | |
| | LeanQuant$_{nu}$ | 2.01 | 65.66 | 32.42 | 54.49 | 66.93 | 23.44 | 25.50 | 23.98 | 28.42 | 45.66 | 72.80 | 66.14 | 45.95 |
| | **Llama-2-13B** | | | | | | | | | | | | | |
| | FP16 | 16 | 79.50 | 48.46 | 70.35 | 76.73 | 42.28 | 47.89 | 61.16 | 59.38 | 60.06 | 79.05 | 72.22 | 63.37 |
| Affine | GPTQ | 4.00 | 78.32 | 45.48 | 68.33 | 75.35 | 40.28 | 46.08 | 56.48 | 54.65 | 58.92 | 78.45 | **71.82** | 59.59 |
| | OmniQuant | 4.00 | 77.69 | 47.10 | 68.74 | 75.57 | 41.39 | 46.10 | 57.39 | 55.87 | **59.48** | **79.00** | 70.32 | 61.70 |
| | LeanQuant$_{aff}$ | 4.00 | **79.42** | **47.27** | **69.16** | **75.90** | **42.21** | **47.31** | **59.90** | **57.93** | 59.07 | 78.24 | **71.82** | **62.57** |
| | GPTQ | 3.00 | 72.85 | 39.85 | 59.77 | 67.20 | 34.86 | 38.85 | 47.97 | 46.48 | 54.61 | 76.28 | 70.32 | 53.62 |
| | OmniQuant | 3.00 | 76.60 | 43.34 | 60.70 | 70.54 | **38.60** | 42.59 | **53.23** | 51.82 | **57.42** | **77.97** | 69.14 | 58.36 |
| | LeanQuant$_{aff}$ | 3.00 | **77.31** | **44.54** | **68.15** | **75.88** | 37.93 | **43.80** | 53.07 | **52.62** | 56.36 | 76.99 | **70.72** | **59.76** |
| | GPTQ | 2.00 | 25.84 | 20.22 | 0.00 | 0.00 | 22.84 | 25.59 | 23.53 | 23.98 | 25.97 | 52.07 | 47.75 | 24.19 |
| | OmniQuant | 2.00 | 48.19 | **24.66** | 10.21 | 20.14 | 21.34 | 24.21 | 21.77 | 23.85 | **40.16** | 63.00 | 52.33 | 31.81 |
| | LeanQuant$_{aff}$ | 2.00 | **50.88** | 24.32 | **32.70** | **39.57** | 21.50 | **24.38** | **21.90** | **24.40** | 38.01 | **67.19** | **56.91** | **36.52** |
| Non-uniform | SqueezeLLM | 4.04 | **78.91** | **47.70** | 70.00 | 76.23 | 42.72 | **47.89** | **60.19** | 58.32 | **59.74** | **78.73** | **72.77** | 63.02 |
| | LeanQuant$_{nu}$ | 4.04 | **78.91** | 47.56 | **71.12** | **77.43** | **43.51** | 47.44 | 59.54 | **58.83** | 59.58 | 78.62 | 72.06 | **63.15** |
| | SqueezeLLM | 3.02 | **77.27** | 43.17 | 66.37 | 73.80 | 38.22 | 44.63 | 55.18 | 53.11 | **58.74** | **77.86** | 69.46 | 59.80 |
| | LeanQuant$_{nu}$ | 3.02 | 77.19 | **44.20** | **71.14** | **78.59** | **40.72** | **45.46** | **56.87** | **55.10** | 56.38 | 77.75 | **70.09** | **61.23** |
| | SqueezeLLM | 2.01 | | | | | - N/A - | | | | | | | |
| | LeanQuant$_{nu}$ | 2.01 | 62.46 | 30.20 | 47.00 | 61.09 | 25.28 | 27.74 | 27.56 | 28.87 | 42.20 | 69.91 | 62.04 | 44.03 |

quantization approaches, such as LLM.int8 (Dettmers et al., 2022a), NormalFloat (Dettmers et al., 2024), and Student Float (Dotzel et al., 2024), perform zero-shot quantization without requiring calibration data. In contrast, methods like GPTQ (Frantar et al., 2022), AWQ (Lin et al., 2024), OmniQuant (Shao et al., 2024), SpQR (Dettmers et al.), SqueezeLLM (Kim et al., 2023), QUIP (Chee et al., 2024), AQLM (Egiazarian et al., 2024), and QUIP# (Tseng et al., 2024) leverage calibration to improve quantization quality by adapting to input data distributions. Some methods (Xiao et al., 2023) extend quantization to both model weights and intermediate activations. Some approaches combine quantization-aware training to push the limits of quantization; for example, LLM-QAT (Liu et al., 2023) fine-tunes quantized models to recover model quality, BitNet (Ma et al., 2024) explores ternary-valued LLMs, while OneBit (Xu et al., 2025) demonstrates the feasibility of 1-bit quantization for LLMs.

Table 11: GPU memory consumption of quantization algorithms on different-sized LLMs. "OOM" means out of memory.

|  | Llama-3-8B | Llama-3-70B | Mistral-Large (123B) | Llama-3.1-405B |
|---|---|---|---|---|
| OmniQuant | 25.3 GB | OOM | OOM | OOM |
| SqueezeLLM | OOM | OOM | OOM | OOM |
| GPTQ | 7.9 GB | 17.1 GB | 32.8 GB | OOM |
| LeanQuant | 7.9 GB | 17.2 GB | 33.0 GB | 65.4 GB |

Table 12: Inference efficiency of 4-bit quantized Llama-3-8B in the decoding and prefill phases. Decoding efficiency is measured in *tokens per second per batch* for generating 4096 tokens, while prefill efficiency is evaluated by *time to first token* for a 4096-token prompt. All results are obtained on an NVIDIA A100-40GB GPU.

|  | **Decoding** (tokens/s per batch) | | | **Prefill** (time to first token) | | |
|---|---|---|---|---|---|---|
|  | Batch Size 1 | Batch Size 4 | Batch Size 16 | Batch Size 1 | Batch Size 4 | Batch Size 16 |
| SqueezeLLM | 26.09 | 19.68 | 7.22 | 153.26s | 333.36s | OOM |
| LeanQuant$_{nu}$ | 23.73 | 18.28 | 9.02 | 0.36s | 1.34s | 5.24s |
| LeanQuant$_{aff}$ | 38.30 | 28.78 | 10.20 | 0.33s | 1.23s | 4.98s |

**Efficient LLMs** Beyond quantization, various techniques have been proposed to enhance LLM efficiency. KV cache compression methods such as KIVI (Liu et al.) and CQ (Zhang et al., 2025c) reduce memory overhead by compressing key-value cache during LLM decoding. Pruning approaches, such as SparseGPT (Frantar & Alistarh, 2023), remove model parameters in a structured or un-structured manner to create sparse, efficient models. Model sketching (Zhang et al., 2025b) enables efficient fine-tuning by compressing LLMs and make them directly fine-tunable. Hardware-aware optimizations, including FlashAttention (Dao et al., 2022) and NoMAD-Attention (Zhang et al., 2025d), improve memory and compute efficiency for modern accelerators. Optimizer-state compression techniques (Zhao et al., 2024; Zhang et al., 2025a) reduce memory usage during pre-training and fine-tuning.

**Uniform and Non-uniform Quantization** Quantization techniques can be broadly categorized into uniform (affine) and non-uniform methods. Uniform quantization (Krishnamoorthi, 2018; Frantar et al., 2022) divides the range of values into equal-sized intervals, which is hardware-efficient but often fails to accommodate the non-uniform distribution of the weights of deep neural networks. Non-uniform quantization improves model compression by allocating precision dynamically based on data distribution. Additive Powers-of-Two Quantization (Li et al.) introduces an efficient non-uniform discretization scheme that leverages power-of-two representations. Nonuniform-to-uniform quantization (Liu et al., 2022) bridges the gap between non-uniform and uniform quantization using a generalized straight-through estimator for training. NUPES (Yvinec et al., 2023) formulates non-uniform post-training quantization as a power exponent search problem. Mr.BiQ (Jeon et al., 2022) focuses on reducing reconstruction error through post-training non-uniform quantization, improving model performance. Methods such as non-uniform step size quantization (Oh et al., 2022) refine quantization granularity to enhance accuracy, while learning-based approaches (Gongyo et al., 2024) adaptively determine step sizes to optimize neural network quantization.

Table 13: Total time taken by LeanQuant for quantizing different-sized LLMs, using a single L40s-48GB GPU, an AMD EPYC 7R13 48-Core CPU, and 370GB of RAM. Llama-3.1-405B is quantized using 2 Quadro RTX 8000 GPUs, an AMD EPYC 7742 64-Core CPU, and 1.48TB of RAM.

| Model | Grid | Group Size | Bits | Time |
|---|---|---|---|---|
| Llama-2-7B | Affine | - | 4.00 | 14 mins |
| | Affine | 128 | 4.25 | 15 mins |
| | Non-uniform | - | 4.05 | 35 mins |
| Llama-3-8B | Affine | - | 4.00 | 16 mins |
| | Affine | 128 | 4.25 | 20 mins |
| | Non-uniform | - | 4.05 | 37 mins |
| Llama-2-70B | Affine | - | 4.00 | 178 mins |
| | Non-uniform | - | 4.04 | 335 mins |
| Mistral-Large-Instruct-2407 (123B) | Affine | - | 4.00 | 252 mins |
| Llama-3.1-405B | Affine | 128 | 4.25 | 1241 mins |

Table 14: The perplexity of LeanQuant models on WikiText2 and C4, using different values of $p$.

| | Grid | Hyperparameter | WikiText2 | | | C4 | | |
|---|---|---|---|---|---|---|---|---|
| | | | 4-bit | 3-bit | 2-bit | 4-bit | 3-bit | 2-bit |
| Mistral-7B | Non-uniform | $p = 0$ | 12.13 | 29.92 | 5,991.18 | 16.73 | 22.71 | 5,998.96 |
| | | $p = 2$ | 5.39 | 5.98 | 25.09 | 7.89 | 8.50 | 20.27 |
| | | $p = 3$ | **5.37** | **5.92** | **22.32** | **7.88** | 8.48 | **19.81** |
| | | $p = 4$ | 5.38 | 5.96 | 25.61 | **7.88** | **8.47** | 21.65 |
| | Affine | $p = 0$ | 14.52 | 80.54 | 230.66 | 16.94 | 69.04 | 243.65 |
| | | $p = 2$ | 5.52 | 8.58 | 55.50 | 8.03 | 16.84 | 41.99 |
| | | $p = 3$ | **5.51** | 6.36 | 18.33 | 8.03 | **8.80** | **20.20** |
| | | $p = 4$ | **5.51** | **6.31** | **18.00** | **8.02** | 8.86 | 20.47 |
| Llama-2-7B | Non-uniform | $p = 0$ | 5.69 | 6.76 | NaN | 7.15 | 8.23 | 62.00 |
| | | $p = 2$ | 5.65 | 6.30 | 17.16 | **7.13** | 7.83 | 19.14 |
| | | $p = 3$ | **5.64** | **6.25** | 17.84 | **7.13** | **7.80** | 19.55 |
| | | $p = 4$ | **5.64** | 6.28 | **15.82** | 7.14 | 7.83 | **18.89** |
| | Affine | $p = 0$ | 5.84 | 8.19 | 93.01 | 7.30 | 9.54 | 85.62 |
| | | $p = 2$ | 5.77 | 7.33 | 27.82 | 7.27 | 8.83 | 28.86 |
| | | $p = 3$ | **5.75** | 6.80 | **25.97** | 7.26 | 8.32 | **27.57** |
| | | $p = 4$ | **5.75** | **6.69** | 26.82 | **7.25** | **8.29** | 28.14 |

Table 15: Ablative experiments on grid point initialization.

| | Grid Init. | Llama-2-7B | | | Llama-3-8B | | | Mistral-7B | | |
|---|---|---|---|---|---|---|---|---|---|---|
| | | 4-bit | 3-bit | 2-bit | 4-bit | 3-bit | 2-bit | 4-bit | 3-bit | 2-bit |
| WikiText2 | K-means++ | **5.64** | 6.25 | 17.84 | **6.59** | 8.31 | 46.31 | **5.37** | 5.92 | 22.32 |
| | Uniformly Spaced (ours) | 5.66 | **6.20** | **17.53** | **6.59** | **7.88** | **41.78** | 5.40 | **5.88** | **19.06** |
| C4 | K-means++ | **7.13** | 7.80 | 19.55 | **10.17** | 12.53 | 39.86 | **7.88** | 8.48 | 19.81 |
| | Uniformly Spaced (ours) | 7.14 | **7.72** | **18.75** | 10.20 | **12.16** | **36.00** | 7.91 | **8.42** | **17.85** |

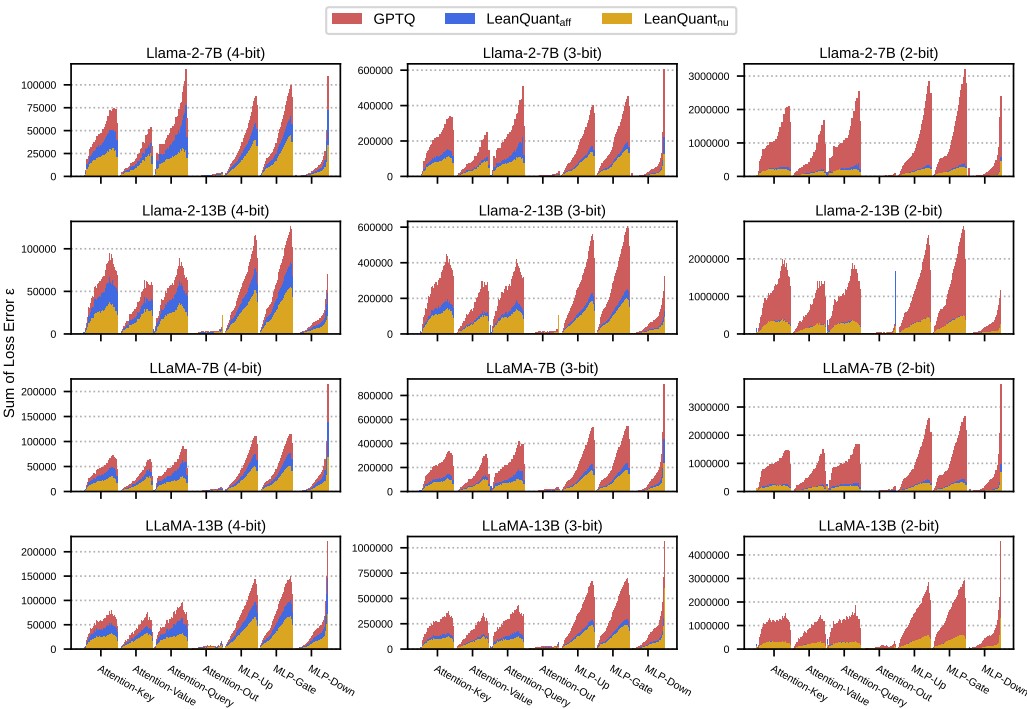

Figure 5: Comparison of loss errors $\epsilon$ of each layer for GPTQ and LeanQuant (affine and non-uniform) during iterative quantization.

