# OpenReview forum: "LeanQuant: Accurate and Scalable Large Language Model Quantization with Loss-error-aware Grid"
_ICLR.cc/2025/Conference — ICLR 2025 Poster_

### Official Review · Reviewer_K8An · 2024-11-02

**Soundness:** 3
**Presentation:** 2
**Contribution:** 3
**Rating:** 5
**Confidence:** 3

**Summary:**

Authors propose “Loss-error-aware Network Quantization” a post-training quantization method for reducing memory requirements in addition to latency of LLMs. The method is proposed as an accurate, versatile, and scalable alternative to traditional methods which are either specific hardware and software platforms or require high resources making them unscalable for the larger models.
The method is based on an iterative loss-error-based quantization framework, overcoming limitation in previous methods: the min-max affine quantization grid fails to preserve model quality due to outliers in inverse Hessian diagonals, instead, authors use loss-error-aware quantization grids, accounting for these outliers.

**Strengths:**

1.	The paper has written “BACKGROUND” sections well, giving an amazing context of quantization.
2.	Results validate the superiority of the proposed quantized method.
3.	The claim of scalability and accuracy is satisfactorily answered,
4.	Algorithm 1 seems to help make sense of the procedure visually.
5.	The ablation study is written well, which helps make the paper a lot more sense, especially Q1, and Q2.

**Weaknesses:**

1)	**The paper assumes the readers are well-versed in quantization**. Two elementary concepts of this work: *quantization grid* and affine / non-uniform grids, are not clearly defined inside the paper.
a.	**What exactly is grid?** [Line 110] “representing the full-precision floating-point parameters with a limited set of grid points on the quantization grid” What does this mean? An example or visualization would be really helpful here.
b.	**[Line 116] “grid points are evenly spaced over the dynamic range of a group of weights”** What does this mean?
c.	A simpler discussion/visualization on uniform and non-uniform grids is requested.
d.	Equation 6, is first solved to establish a quantization grid. It's confusing to know what the solution to equation 6 means if the reader is unfamiliar with the quantization grid.
e.	[Line 117] Group of weights? What’s that? Quantization needs some form of grouping of weights, why? Works don’t seem to establish it.


2)	**Page 6 needs a simpler rewrite if possible**.
a.	The works seem to be indicating some problems with the grouping of weights using KNN, but it's confusing to understand the problem.
b.	[Line 285] equation doesn’t have an equation number
c.	It's not clear what part is already established and what parts are being proposed by the work. Equation 6: where did this equation come from or is derived in some previous work?, similarly Equation 7, and the equation on [Line 285], are authors proposing these equations. There doesn’t seem to be any explanation/derivation for these, just of out blue written as a solution. If they are already established solutions, citing works will help.
d.	“Efficient Fused GPU Kernel for Grid Learning” gives the 50x efficiency. Can this multi-threading technique be applied to existing work and make them more efficient as well?

3)	The solution is based on the presence of outliers with abnormally high values in empirical distributions of inverse Hessian diagonals, (figure 1). This can be dataset-specific or model-specific. Show the presence of outliers in other datasets as well as models as well.

4)	**Contradictory results?** Table 10 shows p values as not so important for quantization. P > 0 is good enough for solving highly accurate quantization? Why does Table 10 doesn’t have p=1? Also, p >1 indicates the “extra” weightage of “outliers” in quantization, the first separation from existing work.
[Line 259] : “The min-max affine grid employed by previous iterative loss-error-based quantization methods do not account for the existence of inverse-diagonal outliers, which may cause significant model quality degradation”
Does this mean, that just by the presence of outliers p=1, has more or less the same performance as P > 1, just accounting for outliers creates better grid formation?
Results for other techniques (SqueezeLLM, GPTQ, OmniQuant) with p=1 or >1 (equation 6), outlier accounted grids can help validate the importance of this outlier driven gird.

5)	Minor issue:
a.	Table 2 & 3: please mention of vanilla FP16 accuracy.
b.	Table 2 & 3: Also non-uniform LeanQuant accuracy.
c.	Table 4: GPTQ Memory values missing

**Questions:**

Please address all the weakness

---

> ### Author Response · Authors · 2024-11-25
> **Response (1 of 2)**
>
> We thank the reviewer for their thoughtful and detailed feedback, as well as their valuable suggestions. Below, we address each weakness and question in detail and are happy to provide further clarification if needed.
>
> > [W1] The paper assumes the readers are well-versed in quantization.
>
> We are sorry for the confusion caused. We will add visualizations in the final paper to illustrate the concept of quantization grid and affine/non-uniform quantization clearly.
>
> > [W1.a] What exactly is a grid?
>
> In the context of quantization, a grid refers to a set of values that represent the possible quantized outputs for the full-precision parameters. This naming is consistent with prior works [1]. To clarify the statement on Line 110, the process of quantization involves assigning each full-precision parameter to its nearest grid point on the quantization grid. For instance, consider a simple 2-bit quantization with grid points $[-1, -0.33, 0.33, 1]$. A floating-point weight of 0.25 would be mapped to 0.33, the closest grid point.
>
> > [W1.b] What does line 116 mean?
>
> This statement refers to the most commonly used affine quantization scheme, where the grid points are distributed uniformly within the range of the weights being quantized. The dynamic range of the weights is defined as $[\min(W), \max(W)]$. For instance, if $\min(W)=-1,\max(W)=1$ with 2-bit quantization, the grid points would be evenly spaced as $[-1, -0.33, 0.33, 1]$.
>
> > [W1.c, W1.d] A simpler discussion/visualization on uniform and non-uniform grids is requested. Explanation on the quantization grid.
>
> We thank the reviewer for bringing this to our attention. We will add figures and examples to the manuscript to illustrate the concepts more clearly.
>
> > [W1.e] Group of Weights
>
> Quantization grids are determined based on the minimum and maximum values within a group of weights. Smaller group sizes result in higher quantization precision because the dynamic range is narrower, allowing for finer granularity in the grid points. The practice of grouping weights is standard in the quantization literature and has been widely adopted in prior works [2,3]. In our paper, we follow this established practice and group weights into blocks of 128 contiguous weights for the results presented in Table 3 and Figure 3.
>
> > [W2] Page 6 needs a simpler rewrite if possible.
>
> We apologize for any confusion caused by the complexity of the current version. We will simplify the language and explanations on page 6 in the final version of the paper to improve readability and clarity.
>
> > [W2.a] The works seem to be indicating some problems with the grouping of weights using KNN.
>
> Thank you for highlighting this point. To clarify, we are not indicating any issues with the grouping of weights using KNN. For $b$-bit non-uniform quantization, we need to learn a set of $2^b$ centroids in the quantization grid. In this context, we propose a learning objective in Equation 6 to preserve the precision of weights corresponding to outlier inverse Hessian diagonals by using k-means clustering.
>
> > [W2.b] The equation on [Line 285] does not have an equation number.
>
> We apologize for this oversight and will add an equation number for clarity in the final version of the paper.
>
> > [W2.c] It's not clear what parts are already established and what parts are being proposed by the work.
>
> We appreciate your feedback on this matter. Equations 6 and 7 are novel contributions of our work. They do not replicate any existing approaches, and they are designed to address specific challenges in preserving the precision of weights for outlier inverse Hessian diagonals.
>
> **Equation 6:** This learning objective minimizes the weighted sum of squared errors between quantized weights and full-precision weights, with weights determined by the inverse Hessian diagonals.
>
> **Equation 7:** While similar to Equation 6 in its goal, it introduces an enumerative search for the optimal parameters due to the discrete nature of affine grids.
>
> > [W2.d] Can this multi-threading technique be applied to existing work and make them more efficient as well?
>
> To the best of our knowledge, existing works do not use a similar approach or formulation for learning quantization grids. Therefore, our fused GPU kernel cannot be directly applied to prior methods. However, we hope that our implementation and multi-threading approach will inspire future advancements in this area.
>
> **References**
>
> [1] van Baalen, Mart, et al. "Gptvq: The blessing of dimensionality for llm quantization." arXiv preprint arXiv:2402.15319 (2024).
>
> [2] Frantar, Elias, et al. "Gptq: Accurate post-training quantization for generative pre-trained transformers." arXiv preprint arXiv:2210.17323 (2022).
>
> [3] Lin, Ji, et al. "AWQ: Activation-aware Weight Quantization for On-Device LLM Compression and Acceleration." Proceedings of Machine Learning and Systems 6 (2024): 87-100.

---

> > ### Author Response · Authors · 2024-11-25
> > **Response (2 of 2)**
> >
> > > [W3] The presence of outliers in inverse Hessian diagonals can be dataset-specific or model-specific.
> >
> > The presence of outlier channels in the activations of LLMs has been well-documented in prior research [3,4]. Given that inverse Hessians are derived by inverting the outer products of these activation tensors, it is expected and consistent with theory that the diagonals of inverse Hessians would also exhibit high-magnitude outliers. To further address the reviewer’s concern, we will expand the investigation to additional datasets and models in the final paper.
> >
> > > [W4] Clarifications on $p$.
> >
> > We conducted additional experiments to analyze the effect of $p \in \\{0, 1, 2, 3, 4\\}$ on the accuracy of MMLU, summarized in the table below. The results show that $p=0$ and $p=1$ are suboptimal for maintaining model quality, with the best performance observed at $p=3$ and $p=4$. To ensure consistency across our experiments, we used $p=4$ in the paper.
> >
> > Regarding the reviewer’s question on $p=1$: while accounting for outliers ($p>0$) significantly improves model quality compared to $p=0$, the additional weightage provided by higher values of $p$ (e.g. $p=3$ or $p=4$) further enhances quantization performance by prioritizing the preservation of outlier precision.
> >
> > Finally, we note that existing methods like SqueezeLLM, GPTQ, or OmniQuant do not employ our proposed loss-error-aware grid. Consequently, these methods cannot directly utilize the parameter $p$, as it is specific to the optimization framework introduced in our work.
> >
> > | Model | Bits | p=0 | p=1 | p=2 | p=3 | p=4 |
> > |---|---|---|---|---|---|---|
> > | Llama-2-7B | 4.05 | 39.21 | 40.17 | 40.83 | 41.18 | **41.85** |
> > |  | 3.02 | 32.48 | 30.91 | 35.38 | **36.27** | 35.63 |
> > |  | 2.01 | 22.95 | 22.9 | 23.02 | 22.89 | **23.24** |
> > | Mistral-7B | 4.05 | 50.68 | 58.26 | 58.45 | **58.52** | **58.52** |
> > |  | 3.02 | 34.64 | 51.15 | 53.53 | **53.82** | 53.04 |
> > |  | 2.01 | 23.62 | 23.09 | 27.02 | 29.89 | **30.09** |
> >
> > > [W5] Minor issue: Accuracy and memory values missing
> >
> > Thank you for pointing out these missing details. Unfortunately, due to GPU resource limitations, we were unable to test the FP16 accuracy for the large models during the discussion period (e.g., Llama 3.1 405B requires 800GB of GPU memory). However, we will strive to provide these numbers and include them in the final version of the paper.
> >
> > We have added the non-uniform LeanQuant accuracy for Table 2 below and are actively working on obtaining the corresponding numbers for Table 3. Additionally, we have updated Table 4 to include the GPTQ memory values, which are presented below.
> >
> > | Model | Method | Bits | Arc Easy | Arc Challenge | LAMBADA Standard | LAMBADA OpenAI | MMLU STEM | MMLU Humanities | MMLU Social | MMLU Other | Average |
> > |---|---|---|---|---|---|---|---|---|---|---|---|
> > | Mistral-Large-Instruct-2407 | GPTQ | 4.00 | 84.60 | 63.99 | 74.38 | 80.52 | 76.31 | 77.23 | 89.31 | 85.23 | 78.95 |
> > |  | $\text{LeanQuant}_\text{aff}$ | 4.00 | 85.14 | 63.99 | 74.99 | 81.14 | **76.56** | 77.32 | 89.21 | **85.87** | 79.28 |
> > |  | $\text{LeanQuant}_\text{nu}$ | 4.05 | **87.67** | **64.59** | **76.63** | **81.51** | 76.50 | **78.00** | **89.35** | 85.68 | **79.99** |
> >
> > | Model | OmniQuant | SqueezeLLM | GPTQ | LeanQuant |
> > |---|---|---|---|---|
> > | Llama-3-8B | 25.3 GB | OOM | 7.9 GB | 7.9 GB |
> > | Llama-3-70B | OOM | OOM | 17.1 GB | 17.2 GB |
> > | Mistral-Large (123B) | OOM | OOM | 32.8 GB | 33.0 GB |
> > | Llama-3.1-405B | OOM | OOM | OOM | 65.4 GB |
> >
> > **References**
> >
> > [3] Lin, Ji, et al. "AWQ: Activation-aware Weight Quantization for On-Device LLM Compression and Acceleration." Proceedings of Machine Learning and Systems 6 (2024): 87-100.
> >
> > [4] Dettmers, Tim, et al. "Gpt3. int8 (): 8-bit matrix multiplication for transformers at scale." Advances in Neural Information Processing Systems 35 (2022): 30318-30332.

---

> ### Comment · Reviewer_K8An · 2024-11-25
> **Request for revised version**
>
> I appreciate the author's effort in resolving all the issues. While I'm going through all the responses, the deadline for uploading the revised version is fast approaching. Is it possible to upload an updated version with all the requested changes?
> It would significantly help to read the revised copy (major rewrite promised) to see if there are any lingering issues.  Most of the issues were regarding non-clarity, which in the current form of the paper is difficult to resolve.
>
> I fear, there isn't much space to incorporate all the figures and explanations into the main paper, retaining most of the issues of non-calrity in the final copy. E.g. W1) Figure with example [-1, -0.33, 0.33, 1] needs to be added, showing the significance of Equation 6. Also, Page 6 re-write. Will the grouping of weights explanation be added in the final copy?

---

> > ### Author Response · Authors · 2024-11-26
> >
> > Thank you for your detailed feedback and for pointing out areas needing improvement. We truly appreciate your insights and the opportunity to refine our paper further.
> >
> > We have updated the latest PDF, with all changes highlighted in blue for your convenience. Specifically, we have addressed the following points:
> >
> > 1. We have rewritten Page 6 to enhance clarity.
> > 2. We have included a detailed explanation of affine and non-uniform quantization grids in Appendix A, along with a visualization to illustrate them more effectively. We have also provided an explanation for the grouping of weights.
> >
> > Due to page limitations, some of the changes could not be included in the main body of the paper. However, the camera-ready version typically allows one additional page, which we plan to utilize to incorporate more explanations and figures as necessary.
> >
> > Please let us know if there are any remaining issues or additional revisions required.

---

> > > ### Comment · Reviewer_K8An · 2024-12-01
> > > **Final Review**
> > >
> > > I appreciate the author's effort in resolving all the issues. Most of my issues are fixed.
> > > I went through the revised paper, and the core issue remains the same: The paper assumes the readers are well-versed in quantization. In my opinion, the paper needs to be rewritten quite a bit, and **I would keep my rating the same**.
> > >
> > > The final camera-ready copy doesn't allow for an extra page, so under the current page restrictions, I do not expect the final copy to be any different.
> > >
> > > Some pointers for final submission would be :
> > > 1) Reduce the white spaces between paragraphs (i.e. phrase the sentences to not have a single word in a new line wasting entire space [line 88, line 35, etc].
> > > 2) When mentioning technical words, describe them in place (Post-training quantization (PTQ) Line 15) no matter how trivial they appear.

---

### Official Review · Reviewer_evsw · 2024-11-03

**Soundness:** 2
**Presentation:** 3
**Contribution:** 2
**Rating:** 5
**Confidence:** 4

**Summary:**

The paper proposes LeanQuant, a novel post-training quantization (PTQ) method that addresses the limitations of existing methods in reducing memory requirements and inference costs of large language models (LLMs). LeanQuant identifies a critical issue in prior PTQ methods, where the min-max affine quantization grid fails to preserve model quality due to outliers in inverse Hessian diagonals. To overcome this, LeanQuant learns loss-error-aware grids instead of using non-adaptive min-max affine grids, enabling accurate and versatile quantization. The method uses an iterative loss-error-based quantization framework, which involves computing the loss-error for each layer, learning a loss-error-aware grid for each layer, and quantizing the model using the learned grids. LeanQuant is evaluated on several large language models, including Llama-3.1 405B, one of the largest open-source LLMs, and achieves high accuracy, comparing favorably to recent competitive baselines. The method is also shown to be scalable, achieving accurate quantization of Llama-3.1 405B using two Quadro RTX 8000-48GB GPUs in 21 hours. LeanQuant's ability to learn loss-error-aware grids enables it to generalize to a wide range of quantization types, including affine and non-uniform quantization, making it a promising solution for reducing the memory requirements and inference costs of LLMs. Overall, LeanQuant provides a novel and effective approach to PTQ, enabling the efficient deployment of large language models in real-world applications.

**Strengths:**

Paper Strengths:

**Compared to existing iterative loss-error-based quantization frameworks**: LeanQuant addresses the critical limitation of min-max affine quantization grids, which fail to preserve model quality due to outliers in inverse Hessian diagonals, by proposing learning loss-error-aware grids.

**Compared to prior methods using non-adaptive min-max affine grids**: LeanQuant's approach produces quantized models that are more accurate and generalizes to a wider range of quantization types, including affine and non-uniform quantization.

**Compared to other methods in terms of scalability**: LeanQuant is scalable, achieving very accurate quantization of Llama-3.1 405B, one of the largest open-source LLMs to date, using two Quadro RTX 8000-48GB GPUs in 21 hours, which is an improvement over other methods.

**Weaknesses:**

Paper Weaknesses:

This paper is limited in novelty. The authors proposed two methods: (1) non-uniform loss-error-aware grid and (2) loss-error-aware affine grid. The first method, the non-uniform loss-error-aware grid, is GPTQ combined with k-means, which is similar to GPTVQ [1]. The loss-error-aware affine grid is equivalent to GPTQ with grid range search, which is implemented in the original GPTQ codebase.
Additionally, they did not compare their work with the most recent state-of-the-art (SoTA) papers.

[1] GPTVQ: The Blessing of Dimensionality for LLM Quantization

**Questions:**

The authors emphasize that range settings are important for quantized network accuracy, and GPTQ uses min-max quantization. In fact, GPTQ employs grid search (instead of SGD) to find the best range setting that minimizes the MSE error (||W-Wq||) https://github.com/IST-DASLab/gptq/blob/main/quant.py#L78-L95 Can authors compare with GPTQ with this MSE range settings? Can the author also explain the conceptual difference between loss-error-aware affine grid and GPTQ + MSE grid search?

Regarding the non-uniform loss-error-aware grid, since the weight value changes during iterative GPTQ quantization, it might affect the k-means centroid as well. Did the authors consider this effect? Did the authors measure how far the weight - \delta deviates from the original weight?

The authors provide two methods: non-uniform quantization and affine quantization. Can the authors briefly explain in which cases each quantization scheme is recommended?

Can authors compare the results with the most recent LLM quantization works, such as QuIP[1], QuIP#[2], QuaRot[3] and SpinQuant[4]?

[1] QuIP: 2-Bit Quantization of Large Language Models With Guarantees

[2] QuIP#: Even Better LLM Quantization with Hadamard Incoherence and Lattice Codebooks

[3] QuaRot: Outlier-Free 4-Bit Inference in Rotated LLMs

[4] SpinQuant: LLM Quantization with Learned Rotations

---

> ### Author Response · Authors · 2024-11-25
> **Response (1 of 2)**
>
> We sincerely appreciate the reviewer’s thoughtful comments and valuable suggestions. Below, we provide detailed responses to the concerns raised.
>
> > **[W1] Novelty of the Paper: Comparison with GPTVQ and GPTQ with Grid Range Search**
>
> Novelty in research is a multifaceted concept, but it can generally be categorized into two key aspects: empirical novelty, which entails uncovering previously unknown properties or phenomena, and technical novelty, which involves proposing new solutions or methods. We argue that our work demonstrates both types of novelty, as outlined below.
>
> **Empirically Novel Observation: High-Magnitude Outliers in the Inverse Hessian Diagonals** We introduce a novel empirical observation: the presence of high-magnitude outliers in the inverse Hessian diagonals, which lead to significant quantization errors and model quality degradation in existing iterative loss-error-based quantization methods. This finding uncovers a previously overlooked aspect of LLM quantization, potentially guiding future advancements.
>
> **Technically Novel Approach: Accurate, Scalable, and Versatile Quantization** Building on this empirical insight, we propose a new quantization method that improves model quality by utilizing loss-error-aware grids. Our approach is *versatile*, as it adapts to both affine and non-uniform quantization formats, and *scalable*, efficiently supporting large-scale models, such as Llama 3.1 405B, with moderate hardware resources.
>
> **Comparison with GPTVQ and GPTQ with Grid Range Search** While GPTVQ extends GPTQ to vector quantization by using multi-dimensional centroids in each codebook, LeanQuant offers broader generalizability. Unlike GPTVQ which is limited to vector quantization, LeanQuant can seamlessly accommodate multiple quantization formats, including both affine and non-uniform grids. This versatility enhances the applicability of the method, making it compatible with popular frameworks such as llama.cpp and TensorRT-LLM, which currently do not support vector quantization.
>
> Additionally, we provide a detailed comparison with GPTQ using grid range search in the response below, and also include comparisons with other recent methods.
>
> > **[Q1] Comparison and Conceptual Differences with GPTQ Using MSE Range Search**
>
> We thank the reviewer for their insightful suggestion. In response, we conducted additional experiments comparing LeanQuant with GPTQ using MSE range search, presented in the table below. LeanQuant achieves superior performance on benchmarks and significantly improved scalability (7x faster quantization).
>
> GPTQ with MSE range search employs an uninformed grid search strategy to minimize the mean squared error (MSE) between full-precision and quantized weights. In contrast, LeanQuant is motivated by the observation of outliers in the inverse Hessian diagonals. By specifically addressing these outliers, LeanQuant effectively mitigates loss errors and preserves model quality. Furthermore, to enable efficient scaling of this loss-error-aware grid learning to large models, we designed and implemented optimized GPU kernels.
>
> | Model | Method | Bits | Quantization Time | Arc Easy | Arc Challenge | LAMBADA Standard | LAMBADA OpenAI | MMLU STEM | MMLU Humanities | MMLU Social | MMLU Other | Average |
> |---|---|---|---|---|---|---|---|---|---|---|---|---|
> | Llama-2-7B | GPTQ with MSE range search | 4.00 | 105 mins | 72.39 | 40.78 | 63.65 | 70.52 | 32.79 | 31.54 | 40.92 | 38.69 | 48.91 |
> |  | $\text{LeanQuant}_\text{aff}$ | 4.00 | 14 mins | **75.00** | **41.21** | **65.03** | **72.02** | **34.82** | **36.94** | **46.77** | **44.54** | **52.04** |
> |  | GPTQ with MSE range search | 3.00 | 111 mins | 71.63 | 38.14 | 57.07 | 67.92 | 30.45 | 29.33 | 35.42 | 33.54 | 45.44 |
> |  | $\text{LeanQuant}_\text{aff}$ | 3.00 | 14 mins | **71.84** | **38.99** | **59.13** | **69.05** | **33.56** | **32.96** | **41.11** | **40.10** | **48.34** |
> |  | GPTQ with MSE range search | 2.00 | 103 mins | 35.77 | **20.99** | 5.86 | 9.08 | **21.44** | 23.25 | 21.74 | 23.01 | 20.14 |
> |  | $\text{LeanQuant}_\text{aff}$ | 2.00 | 14 mins | **41.08** | **20.99** | **16.98** | **21.93** | 21.25 | **24.06** | **21.77** | **23.88** | **23.99** |
>
> > **[Q2] Did the authors consider the effect of weight value changes during iterative quantization on the k-means centroids?**
>
> We thank the reviewer for raising this insightful point. Yes, we have carefully considered this effect and provided a discussion in Section B of the Appendix to explain why it does not pose an issue in practice.

---

> > ### Author Response · Authors · 2024-11-25
> > **Response (2 of 2)**
> >
> > > **[Q3] Can the authors briefly explain in which cases each quantization scheme is recommended?**
> >
> > Non-uniform quantization is recommended when achieving high accuracy is the primary goal. This scheme generally outperforms affine quantization in terms of preserving model quality under comparable model sizes, making it ideal for applications where accuracy is critical.
> >
> > In contrast, affine quantization is the most widely supported format across existing hardware and software frameworks, offering significant advantages in terms of compatibility and deployment. Additionally, it provides lower inference latency due to its simplicity and efficient implementation on standard hardware. As a result, affine quantization is better suited for scenarios where speed or broad compatibility is a top priority.
> >
> > > **[Q4] Comparison with More Recent LLM Quantization Works**
> >
> > We conducted additional experiments to compare LeanQuant with QuaRot, using `lm-eval version 0.4.0` for all evaluations, as it is the only version supported by the QuaRot codebase. Our results demonstrate that LeanQuant performs either better or on par with QuaRot in terms of accuracy.
> >
> > Additionally, we note that QuaRot can be combined with LeanQuant to further enhance performance. Since QuaRot's accuracy depends on the underlying quantization algorithm (RTN or GPTQ), using LeanQuant as the base algorithm for QuaRot has the potential to improve quantization accuracy even further.
> >
> > It is also important to highlight that QuaRot applies Hadamard transforms in the resulting quantized model, which introduces additional inference overhead. As a result, QuaRot models may have slower inference times compared to LeanQuant models, making LeanQuant a more efficient choice in latency-sensitive scenarios.
> >
> > | Model | Method | Bits | Arc Easy | Arc Challenge | LAMBADA Standard | LAMBADA OpenAI | MMLU STEM | MMLU Humanities | MMLU Social | MMLU Other | Average |
> > |---|---|---|---|---|---|---|---|---|---|---|---|
> > | Llama-3-8B | QuaRot-RTN | 4.00 | 65.92 | 46.93 | 65.75 | 71.92 | 49.67 | 51.24 | 65.75 | 64.24 | 60.18 |
> > |  | QuaRot-GPTQ | 4.00 | **76.81** | 50.43 | 67.53 | 74.19 | **51.60** | 53.28 | 70.26 | 68.23 | 64.04 |
> > |  | $\text{LeanQuant}_\text{aff}$ | 4.00 | 76.30 | **50.51** | **68.27** | **75.90** | 51.51 | **53.43** | **70.69** | **68.43** | **64.38** |
> > | Llama-2-13B | QuaRot-RTN | 4.00 | 74.54 | 47.44 | 66.19 | 74.17 | 39.74 | 41.74 | 53.04 | 51.95 | 56.10 |
> > |  | QuaRot-GPTQ | 4.00 | 76.35 | **49.23** | 69.82 | **76.48** | 41.29 | 47.01 | 58.99 | 57.84 | 59.63 |
> > |  | $\text{LeanQuant}_\text{aff}$ | 4.00 | **76.43** | 49.06 | **69.90** | **76.48** | 42.53 | **47.86** | **60.03** | **58.93** | **60.15** |

---

> > > ### Comment · Reviewer_evsw · 2024-11-26
> > >
> > > Thank you for your response. However, I still have some concerns regarding the limited novelty of the paper, so I have decided to keep my original score.

---

### Official Review · Reviewer_uLwi · 2024-11-03

**Soundness:** 2
**Presentation:** 2
**Contribution:** 2
**Rating:** 3
**Confidence:** 3

**Summary:**

The paper introduces a new quantization method, LeanQuant, designed to improve the efficiency and accuracy of large language models (LLMs) without requiring specialized hardware adaptations.

Key Contributions:
1. LeanQuant innovates on post-training quantization (PTQ) by introducing a loss-error-aware quantization grid. This grid preserves model quality better than traditional min-max grids for outliers in the inverse Hessian diagonals, which otherwise lead to significant accuracy loss.
2. The method generalizes to affine and non-uniform quantization types, making it adaptable to existing frameworks and more compatible with standard inference kernels.
3. LeanQuant demonstrates scalability with successful quantization of models up to 405 billion parameters, achieving high accuracy using limited computational resources.
4. By designing a fused GPU kernel, LeanQuant achieves substantial speedups in grid learning, making quantization efficient and suitable for a range of model sizes.

**Strengths:**

The strength of this paper lies in its innovative approach to addressing both accuracy and scalability in quantizing large language models, all while maintaining low hardware requirements. LeanQuant introduces a loss-error-aware quantization grid,  tackling the common issue in traditional quantization methods where accuracy is lost due to outliers in inverse Hessian diagonal elements. This approach adapts the quantization grid to preserve the precision of key weights, improving model quality, especially in low-bit quantization.

Additionally, LeanQuant’s multi-format compatibility enables it to integrate seamlessly with popular deep learning frameworks, eliminating the need for specialized inference kernels and greatly enhancing deployment flexibility.

On the scalability side, LeanQuant incorporates a fused GPU kernel design, accelerating the quantization process and allowing it to efficiently handle extremely large models (up to 405 billion parameters) with limited GPU resources. This efficiency showcases LeanQuant’s practicality under constrained computing resources, highlighting its potential for low-cost, high-efficiency AI applications. As a result, the paper presents a robust solution for large-scale model quantization, excelling in quantization quality, resource efficiency, and deployment adaptability.

**Weaknesses:**

1.  Quantization isn't solely about reducing memory usage or data transfer—it’s fundamentally about reducing the computational workload, which in turn decreases the size of intermediate data and the need for data movement. If the focus is only on reducing model size without addressing the computational load, it could result in diminished acceleration benefits. This is because, even if memory requirements decrease, hardware may still need to perform the same level of computation, which undermines the intended efficiency.
2. As you pointed out, while LeanQuant can quantize models down to 3-bit or 2-bit, modern GPUs often support 4-bit or 8-bit multiplications. Thus, quantizing to lower bit widths (like 3 or 2 bits) might not yield computational efficiency if, during execution, the kernel still calls for a 4-bit multiplier. This forced conversion undermines the benefits of lower-bit quantization since it fails to leverage the intended reduction in computational and resource demands.
3. Without considering the storage format of hardware, 3-bit or 2-bit weights may still be stored on the GPU in higher bit-widths (e.g., 4-bit, 8-bit, or even 16-bit), resulting in excess storage and data transfer requirements. This dilutes the benefits of quantization. Designing quantization methods with hardware-oriented data formats is essential to ensure that low-bit quantized data can be efficiently processed and stored by hardware.

**Questions:**

The author’s motivation is well-founded, and the experiments are thorough. However, addressing the issues I raised in the weaknesses would greatly strengthen the paper.

---

> ### Author Response · Authors · 2024-11-25
>
> We appreciate the reviewer’s thoughtful feedback and recognition of our contributions to accuracy and scalability in LLM quantization. Below, we address the concerns raised and provide additional evidence. We hope these responses clarify the issues and encourage reconsideration of the score.
>
> > [W1] Quantization isn't solely about reducing memory usage or data transfer
>
> Thank you for raising this point. While quantization does not inherently reduce the computational workload, it significantly accelerates LLM inference due to two key factors:
>
> **LLM Inference as an IO-Bound Operation:** As highlighted in [1], LLM inference is primarily IO-bound rather than compute-bound. This means that loading weights into memory, rather than performing computations, is often the primary bottleneck for latency. By significantly reducing the size of weights, quantization alleviates this bottleneck and accelerates inference.
>
> **Reduced Communication Overhead in Multi-GPU Settings:** When an LLM is split across multiple GPUs, quantization reduces the number of GPUs required for inference. This reduces the communication overhead between GPUs, further improving efficiency and scalability.
>
> Existing research, such as [2], demonstrates that quantized models can achieve substantial speedups (2-4x) over full-precision models. Additionally, we have conducted new experiments benchmarking token throughput and peak memory usage for 2-bit, 3-bit, and 4-bit affine LeanQuant models for decoding 1K tokens using the exllamav2 kernel on an L40s GPU. The results clearly show that LeanQuant models achieve significant speedups over FP16 models, further substantiating the practical benefits of our method.
>
> | Model | Method | Tokens Per Second | Peak GPU Memory |
> |---|---|---|---|
> | Llama-2-7B | FP16 | 22.9 | 14.1 GB |
> |  | 4-bit LeanQuant | 32.7 | 4.6 GB |
> |  | 3-bit LeanQuant | 33.2 | 3.8 GB |
> |  | 2-bit LeanQuant | 42.8 | 3.4 GB |
>
> > [W2] Quantizing to lower bit widths (like 3 or 2 bits) might not yield computational efficiency
>
> We respectfully disagree. While it is true that packed low-bit integers are converted to floating-point numbers for multiplications, this conversion is performed directly within GPU kernels with negligible overhead. Many libraries such as exllamav2, llama.cpp, and Marlin use optimized fused GPU kernels to handle these operations efficiently.
>
> Additionally, LLM inference is primarily IO-bound rather than compute-bound, as noted in prior works like [1]. The main bottleneck is loading weights from memory, and quantization significantly reduces global memory IO operations, leading to faster inference. As shown in the table above, LeanQuant achieves substantial improvements in both token throughput and memory usage for 3-bit and 2-bit models, by leveraging the fused kernel in exllamav2.
>
> > [W3] 3-bit or 2-bit weights may still be stored on the GPU in higher bit-widths
>
> We respectfully disagree with this assertion. In practice, 3-bit and 2-bit weights are tightly packed in GPU memory, ensuring reduced memory usage without the overhead of higher bit-width storage. Many LLM inference libraries, such as exllamav2 and llama.cpp, employ efficient packing strategies to store low-bit weights compactly in GPU RAM. This efficiency gain is evident in our experimental results, as shown in the table above, where 2-bit and 3-bit LeanQuant models achieve significant memory savings compared to 4-bit models.
>
> **References**
>
> [1] Kim, Sehoon, et al. "SqueezeLLM: Dense-and-Sparse Quantization." Forty-first International Conference on Machine Learning.
>
> [2] Guo, Han, et al. "Fast Matrix Multiplications for Lookup Table-Quantized LLMs." Findings of the Association for Computational Linguistics: EMNLP 2024. 2024.

---

### Official Review · Reviewer_VgEF · 2024-11-03

**Soundness:** 3
**Presentation:** 3
**Contribution:** 3
**Rating:** 6
**Confidence:** 4

**Summary:**

This paper proposes Loss-error-aware Network Quantization (LeanQuant), a method designed to learn loss-error-aware quantization grids to reduce quantization error. LeanQuant determines optimal quantization parameters, such as scaling factor and zero-point, by iteratively narrowing the weight range in discrete steps. To accelerate grid learning, custom CUDA kernels were developed for efficient grid searching. Consequently, the paper demonstrates that LeanQuant can achieve superior linguistic capabilities with quantized LLMs compared to previous methods.

**Strengths:**

1.	LeanQuant calibrates quantization grids, and it is a novel approach in the post-training quantization of LLMs.
2.	The development of fused CUDA kernels efficiently supports the calibration process, so it makes the complex grid learning procedure more accessible for broad application due to its enhanced efficiency.

**Weaknesses:**

Overall, several details and evaluation results are missing, which makes it difficult to fully understand and be convinced of the effectiveness of LeanQuant.

1.	A detailed explanation of how LeanQuant can be extended to non-uniform quantization is missing.
2.	Although Section 4 (Experiments) states that the proposed method will be compared with AWQ, GPTQ, and OmniQuant, the evaluation results for AWQ are not presented in the main text (some results for AWQ are included in the appendix only).
3.	While group-wise quantization is widely used in LLM quantization, Table 1 appears to evaluate affine quantization only with row-wise quantization. (Note that AWQ is specialized for group-wise quantization, and layer-wise quantization with AWQ sometimes results in slightly worse linguistic capabilities compared to GPTQ.)
4.	The runtime for LeanQuant is presented with and without the custom kernel in Table 5, but no comparison is made with previous methods.
5.	Peak GPU memory usage for AWQ and GPTQ is not included in Table 4.

**Questions:**

1.	Could you provide a step-by-step explanation of how LeanQuant is adapted for non-uniform quantization by highlighting key differences from the affine quantization process?
2.	To ensure consistency between the promised and delivered comparisons, I suggest the authors either include a summary of the AWQ results in the main text or explicitly state why they were omitted.
3.	Why did you choose row-wise quantization for the comparison in Table 1? Additionally, do you have results for group-wise quantization? Providing more analysis with various quantization methods could address potential concerns about the fairness of the comparison and offer insights into LeanQuant's performance across different quantization schemes.
4.	I recommend including runtime comparisons with GPTQ, AWQ, and OmniQuant in Table 5 to provide a more comprehensive view of LeanQuant's efficiency relative to other existing methods
5.	I recommend including peak GPU memory usage for AWQ and GPTQ in Table 4, or explaining why this information was omitted. This would offer a more complete comparison of resource requirements across methods.
6.	How are the quantization grids calibrated for non-uniform quantization? Since each grid is independent for non-uniform quantization, I assume the calibration process must differ from the loss-error-aware affine grid described in Section 3.2.2.
7.	What would happen if group-wise quantization were applied to the affine quantization method presented in Table 1?

---

> ### Author Response · Authors · 2024-11-25
>
> We appreciate the reviewer's thorough review and constructive feedback. Below, we carefully address your concerns.
>
> > [W1,Q1,Q6] How is LeanQuant applied to non-uniform quantization, and what are the key differences from affine quantization?
>
> Thank you for this insightful question. LeanQuant solves a layer-wise optimization problem (Equation 1). Before applying iterative loss-error-based quantization to each layer, the quantization grid is learned and fixed.
>
> The input for learning the quantization grid is the same for both non-uniform and affine quantization: a group of model weights, their corresponding inverse Hessian diagonals, and the parameter $p$. However, the key difference lies in the output and the learning process:
>
> - ***Output Differences:*** **Non-uniform Quantization:** The output is a set of centroids, which are learned through k-means clustering (Equation 6). **Affine Quantization:** The output consists of a scaling factor and a zero-point, learned using enumerative search (Equation 7).
>
> - ***Learning Objective:*** In both cases, the objective is to minimize the weighted sum of errors of the quantized weights, where the weights are scaled by the inverse Hessian diagonals. This ensures that the quantization process prioritizes preserving weights that are more important to model accuracy.
>
> - ***Learning Methodology:*** Non-uniform quantization uses k-means clustering, which allows centroids to move freely to minimize the objective. Affine quantization employs enumerative search, as the grid points in affine quantization must be evenly spaced, making the process inherently discrete.
>
> These differences in grid learning are detailed in lines 5–9 and 12–16 of Algorithm 1, which outline the distinct procedures for non-uniform and affine quantization in LeanQuant.
>
> > [W2,W3,Q2,Q3,Q7] Evaluations on Group-Wise Quantization and Comparison with AWQ
>
> Thank you for highlighting the need for a more detailed evaluation of group-wise quantization and comparisons with AWQ. To address this, we conducted additional experiments on group-wise quantization, including a comparison with AWQ and OmniQuant. The Llama-2 OmniQuant models were obtained from the official repository, while the Llama-3.1 AWQ model was obtained from the official `hugging-quants` community repository.
>
> The results, summarized in the table below, demonstrate that LeanQuant performs better on average across benchmarks than OmniQuant and AWQ in the group-wise quantization setting. These findings further validate the effectiveness and versatility of LeanQuant. We also provide more evaluations of group-wise quantization in Table 3 and Figure 3 of the paper.
>
> | Model | Method | Group Size | Bits | Arc Easy | Arc Challenge | LAMBADA Standard | LAMBADA OpenAI | MMLU STEM | MMLU Humanities | MMLU Social | MMLU Other |  |
> |---|---|---|---|---|---|---|---|---|---|---|---|---|
> | Llama-2-7B | OmniQuant | 128 | 4.25 | 75.21 | **43.69** | 66.95 | 72.91 | 35.11 | 37.79 | 46.77 | 46.86 | 53.16 |
> |  | AWQ | 128 | 4.25 | 75.17 | 43.26 | 67.40 | 72.70 | 34.89 | 37.79 | 46.34 | 45.93 | 52.94 |
> |  | LeanQuant | 128 | 4.25 | **76.26** | 43.00 | **67.75** | **74.44** | **35.65** | **39.02** | **47.94** | **49.79** | **54.23** |
> | Llama-2-13B | OmniQuant | 128 | 4.25 | 78.24 | **47.61** | 69.75 | 75.99 | **42.06** | 47.74 | **59.83** | 58.61 | 59.98 |
> |  | AWQ | 128 | 4.25 | 78.91 | 46.50 | 70.17 | 76.19 | 41.39 | 46.70 | 59.38 | 55.94 | 59.40 |
> |  | LeanQuant | 128 | 4.25 | **79.00** | 47.18 | **70.75** | **77.95** | 41.96 | **47.86** | 59.54 | **58.97** | **60.40** |
> | Llama-3.1-70B-Instruct | AWQ | 128 | 4.25 | **86.62** | **62.12** | 72.09 | 75.70 | 74.88 | 80.60 | 87.07 | **83.71** | 77.85 |
> |  | LeanQuant | 128 | 4.25 | 86.49 | 61.69 | **72.70** | **76.07** | **75.61** | **80.79** | **87.55** | **83.71** | **78.08** |
>
> > [W4,W5,Q4,Q5] Runtime and Memory Usage Comparison
>
> Thank you for your suggestion to provide a more detailed comparison of runtime and memory usage. Below, we present additional benchmarking results comparing LeanQuant with AWQ and GPTQ. For AWQ and GPTQ, we used the official AutoAWQ and AutoGPTQ packages.
>
> | Model | Method | Peak GPU Memory | Quantization Time |
> |---|---|---|---|
> | Llama-3-8B | AWQ | 19.4 GB | 17 mins |
> |  | GPTQ | 7.9 GB | 10 mins |
> |  | LeanQuant | 7.9 GB | 16 mins |
> | Llama-3-70B | AWQ | OOM | - |
> |  | GPTQ | 17.1 GB | 97 mins |
> |  | LeanQuant | 17.2 GB | 185 mins |
>
> The results indicate that LeanQuant is more memory-efficient and faster than AWQ, but less efficient in runtime compared to GPTQ. However, LeanQuant achieves higher accuracy than GPTQ, offering a trade-off between efficiency and precision. Although LeanQuant is slower than GPTQ, it remains highly scalable, as demonstrated by its ability to quantize one of the largest open-source LLMs to date (405B parameters). We will provide a more comprehensive comparison on memory usage and quantization time in the final paper.

---

> > ### Comment · Reviewer_VgEF · 2024-11-26
> >
> > Thank you for the detailed response. Your explanation has addressed my concerns, and I have raised my score to 6.

---

### Official Review · Reviewer_v8KG · 2024-11-03

**Soundness:** 3
**Presentation:** 3
**Contribution:** 3
**Rating:** 6
**Confidence:** 4

**Summary:**

The paper proposes LeanQuant, an accurate, versatile, and scalable quantization approach. LeanQuant is built upon the iterative loss-error-based quantization framework and identify one of the biggest limitations of such methods: the quantization grid introduces high quantization errors due to the existence of outliers in the inverse Hessian diagonals.

**Strengths:**

The idea is simple and effective, compatible with iterative loss-error-based quantization approaches such as GPTQ. The calibration process is relatively lightweight, allowing for the quantization of models up to 405B. The experimental design of this paper considers the latest model series and different model sizes, which is comprehensive.

**Weaknesses:**

LeanQuant is built on the core idea of using "inverse Hessian diagonals" metric to quantify the importance of weights. Similar ideas are also introduced in works such as SqueezeLLM [1], which exploits the "Hessian diagonal" metric directly derived from the quantization loss function to identify important weights. Compared to the metric in SqueezeLLM, I think the derivation of the metric in LeanQuant is relatively heuristic. Although LeanQuant beats SqueezeLLM in the final performance, it would be better to directly compare the different importance metrics and show the strengths of the proposed "inverse Hessian diagonals" metric since SqueezeLLM does not leverage GPTQ to minimize the quantization error.

[1] Kim, Sehoon, et al. "Squeezellm: Dense-and-sparse quantization." arXiv preprint arXiv:2306.07629 (2023).

**Questions:**

1. Whether it can be compared or ablated with other importance metrics such as the "Hessian diagonal" in SqueezeLLM, these works are similar in idea. Please see the weakness part.

2. There is a lack of literature survey on uniform and non-uniform quantization in related works.

---

> ### Author Response · Authors · 2024-11-25
>
> We thank the reviewer for recognizing the strengths of LeanQuant and its comprehensive experimental design. We appreciate your insightful feedback and suggestions and address your concerns and questions in detail below.
>
> > [W1] Similar ideas of using "inverse Hessian diagonals" have been introduced in other works.
>
> Thank you for pointing this out. While both LeanQuant and SqueezeLLM employ Hessian-based metrics to quantify the importance of weights, there are key distinctions in the formulation and application of these metrics:
>
> **Fundamental Difference in Metrics:** SqueezeLLM uses the diagonal of the Fisher Information Matrix, derived from the squared gradient of the network loss. In contrast, LeanQuant leverages the inverse Hessian diagonals of the layer-wise optimization objective (Equation 1), computed directly from activations, eliminating the need for gradient calculations.
>
> **Scalability Advantage:** The metric used by SqueezeLLM requires gradients computed across the entire model, making it resource-intensive, especially for large models. In contrast, LeanQuant decomposes quantization into a layer-wise optimization problem and calculates the Hessian using only the activations of each layer. This design dramatically improves scalability. For instance, as demonstrated in Table 4 of our paper, SqueezeLLM fails to quantize an 8B parameter model on a 48GB GPU, while LeanQuant successfully quantizes a 123B parameter model under the same constraints.
>
> **Comparison of Metrics:** To address your suggestion, we conducted experiments comparing the performance of LeanQuant with the SqueezeLLM metric for learning the quantization grid, shown in the table below.
>
> > [Q1] Ablation with Other Importance Metrics such as the "Hessian Diagonal" in SqueezeLLM
>
> We appreciate the reviewer’s suggestion to include ablations with other importance metrics. In the table below, we present additional experiments comparing the performance of LeanQuant’s metric with the "Hessian diagonal" metric used in SqueezeLLM for learning the quantization grid. The results show that LeanQuant’s metric outperforms SqueezeLLM’s metric in 3-bit and 2-bit settings while performing comparably in the 4-bit region.
>
> Additionally, we highlight key advantages of LeanQuant in terms of scalability and versatility:
>
> **Scalability:** SqueezeLLM is memory-intensive, requiring approximately 1.6 TB of GPU memory to quantize a 405B parameter model. In contrast, LeanQuant quantizes the same model using just 65.4 GB of GPU memory, making it far more practical for large-scale models.
>
> **Versatility:** SqueezeLLM is limited to non-uniform quantization. LeanQuant, however, supports multiple quantization formats, including affine and non-uniform quantization, and performs favorably against competitive baselines in both cases.
>
> | Model | Method | Importance Metric | Bits | Arc Easy | Arc Challenge | LAMBADA Standard | LAMBADA OpenAI | MMLU STEM | MMLU Humanities | MMLU Social | MMLU Other | Average |
> |---|---|---|---|---|---|---|---|---|---|---|---|---|
> | Llama-3-8B | LeanQuant | SqueezeLLM | 4.05 | **80.01** | 48.03 | **67.78** | **75.99** | **52.52** | **53.86** | 70.62 | 68.45 | **64.66** |
> |  | LeanQuant | LeanQuant | 4.05 | 79.50 | **49.15** | 67.36 | 74.95 | 52.17 | 53.16 | **71.40** | **68.75** | 64.56 |
> |  | LeanQuant | SqueezeLLM | 3.02 | 76.56 | 43.60 | 62.88 | 71.67 | 45.23 | 47.44 | 61.33 | 59.45 | 58.52 |
> |  | LeanQuant | LeanQuant | 3.02 | **77.74** | **47.01** | **63.32** | **72.17** | **48.84** | **49.05** | **65.45** | **62.79** | **60.80** |
> |  | LeanQuant | SqueezeLLM | 2.01 | 39.10 | 19.20 | 26.02 | 27.69 | 22.23 | 24.70 | 22.68 | 25.14 | 25.85 |
> |  | LeanQuant | LeanQuant | 2.01 | **58.21** | **26.62** | **31.22** | **39.16** | **25.98** | **25.48** | **27.01** | **26.65** | **32.54** |
>
> > [Q2] Lack of Literature Survey on Uniform and Non-Uniform Quantization in Related Works
>
> We thank the reviewer for bringing this to our attention. We appreciate the reviewer’s suggestion and will include a comprehensive literature survey on uniform and non-uniform quantization in the final version of the paper.

---

> > ### Comment · Reviewer_v8KG · 2024-11-25
> >
> > Thanks for your reply, which addressed my key concerns. I decide to raise my score to 6. Congrats!

---

### Official Review · Reviewer_zNzV · 2024-11-03

**Soundness:** 4
**Presentation:** 3
**Contribution:** 3
**Rating:** 6
**Confidence:** 4

**Summary:**

This paper presents a method for optimizing the quantization grid in a layerwise loss-aware manner to reduce low-bit weight quantization error. Unlike previous optimization-based approaches like GPTQ, they derive expressions for the optimal settings for the quantization grid rather than the optimal weight update to apply at each step. The approach is generalizable to both non-uniform and uniform quantization grids. This method is able to preferentially preserve more sensitive values, while maintaining computational efficiency by leveraging the layerwise optimization objective. This grid-tuning approach can also be integrated with previous weight update approaches like GPTQ to attain higher accuracy. They demonstrate the accuracy benefits of their approach, and also highlight the computational and memory efficiency of applying their quantization method relative to prior works.

**Strengths:**

- They present an algorithm to tune the quantization grid using the layerwise loss-aware objective
- They present both non-uniform and uniform methods (the non-uniform method is flexible and leverages clustering, whereas the non-uniform method performs a constrained search over potential scale factors)
- They provide accelerated GPU kernels to solve the objective for their affine quantization approach
- They provide detailed analysis of their method against prior work in both uniform and non-uniform quantization regimes
- They provide multiple ablations demonstrating how their method reduces the loss error, as well as the impacts of K-means initialization

**Weaknesses:**

- Although the method of determining weight sensitivity using the layerwise loss is distinct, the approach of performing K-Means clustering in equation (6) to derive non-uniform datatypes is the same as prior work (eg. the sensitivity-weighted clustering approach to derive non-uniform datatypes in SqueezeLLM)

**Questions:**

- Can you provide any intuition for why the uniform grid point spacing method helps improve accuracy for k-means than k-means++ style grid points (i.e. a visualization of where the initial grid points end up being placed in both cases, versus the “target” grid points)?
- For the efficiency results in Table 9, is the K-means non-uniform quantization grid search parallelizable on the GPU, or is it being run on the CPU?

---

> ### Author Response · Authors · 2024-11-25
>
> We thank the reviewer for carefully reviewing our paper and providing insightful feedback. Below, we address the concerns and questions raised.
>
> > [W1] Similarity to SqueezeLLM in using K-Means Clustering for Non-Uniform Datatypes
>
> While the K-means clustering approach in LeanQuant shares some similarities with the sensitivity-weighted clustering in SqueezeLLM, there are key differences that we would like to emphasize:
>
> **Motivation from Inverse Hessian Outliers:** LeanQuant is specifically motivated by the outliers in inverse Hessian diagonals, a characteristic unique to iterative loss-error-based quantization approaches. This focus ensures better preservation of weights that may introduce high loss errors during quantization.
>
> **Scalability:** LeanQuant is designed to be far more scalable than SqueezeLLM. By leveraging layer-wise optimization, LeanQuant can quantize models with up to 123B parameters on a single 48GB GPU, whereas SqueezeLLM fails to quantize an 8B model.
>
> **Versatility:** LeanQuant is applicable to a wider range of quantization formats, including both affine and non-uniform quantization, while SqueezeLLM is limited to non-uniform formats.
>
> > [Q1] Intuition for Why Uniform Grid Point Spacing Improves Accuracy
>
> Thank you for this thoughtful question. The main intuition lies in the distribution of weights:
>
> **Weight Distribution Characteristics:** Weights tend to be densely distributed near the center and sparsely distributed at the extremes. Standard initializations such as random and k-means++ tend to undersample these extreme values, which may fail to adequately represent these sparsely populated regions.
>
> **Uniform Grid Initialization:** By initializing the centroids uniformly across the range of minimum and maximum weights, LeanQuant ensures that extreme values are adequately represented in the quantization grid without compromising the precision of non-extreme values.
>
> We plan to include illustrations in the final paper to further clarify this process.
>
> > [Q2] Efficiency of K-Means Non-Uniform Quantization Grid Search
>
> The K-means grid search process reported in Table 9 was conducted on the CPU. However, this process is fully parallelizable on the GPU, which would enable significant speedups. We would like to adopt this approach in the near future.

---

> > ### Comment · Reviewer_zNzV · 2024-11-26
> > **Response**
> >
> > Thank you to the authors for the clarifications around the differences between their method and prior work, as well as the intuition for the benefits of uniform grid initialization and around the details of the k-means method. I will retain my initial score.

---

### Meta-Review · Area_Chair_P9wd · 2024-12-23

**Metareview:**

This paper proposes LeanQuant, a post-training quantization (PTQ) method to mitigate the memory and inference cost challenges in LLMs. One of the main contributions of the paper is the introduction of loss-error quantization grids, which adapt to outliers in inverse Hessian diagonals. Outliers are known to significantly affect quantization quality. The paper claims to be generalizable to both affine and non-uniform quantization grids and shows compatibility with standard inference kernels (no hardware specific changes). With respect to results, the paper reports strong experimental results across different LLMs, models up to 405B params. LeanQuant outperforms SOTA methods like SqueezeLLM and GPTQ in both scalability and accuracy, esp. for low-bit quantization.

These are the main strengths of the paper: (a) the introduction of loss-error-aware quantization grids (works for both affine and non-uniform quantization) to address outlier sensitivity, improving quantization quality, esp. in low-bit representation regime. (b) showing results for up to 405B params on modest hardware setups (48GB GPUs), outperforming SOTA methods like SqueezeLLM. (c) using fused GPU kernels to accelerate grid learning, yielding speedups and enabling the quantization of large models.  (d) extensive experimentations on different models (Llama-2, Llama-3, and Mistral).

There are multiple weaknesses about the paper as well: (a) there seem to be some similarities between LeanQuant and prior methods like GPTQ and SqueezeLLM. For instance, LeanQuant's non-uniform grid learning via k-means clustering shares conceptual overlap with SqueezeLLM. In addition, one reviewer also mentioned that the introduced affine quantization component is equivalent to GPTQ w/ grid range search. (b) some results are missing. esp. comparison with SOTA methods like QuaRot, QuIP and SpinQuant in the main text. (c) some reviewers talked about the fact that lower-bit quantization (3-bit and lower) might not be relevant with current hardware support (but I believe this is true for any low-bit quantization methods).

This is a borderline paper and most reviewers are positive about this work. One reviewer criticized the paper on writing that I don't believe should be the main reason for rejection. The other reviewer with most negative score also didn't engage in the discussion. After reading the rebuttal, I think the authors did a good job addressing the reviewer's concern. Overall, based on the reviewers' assessment and my reading, I recommend **Accept** for this paper.

**Additional Comments On Reviewer Discussion:**

Reviewers raised several points regarding clarity, novelty, comparisons with prior work, and the practical implications of the proposed method.

(a) (Reviewer zNzV) questioned the novelty of LeanQuant, highlighting its conceptual overlap with existing methods such as GPTQ and SqueezeLLM. For instance, LeanQuant’s k-means-based non-uniform quantization grid learning shares similarities with sensitivity-weighted clustering in SqueezeLLM, and its affine quantization is likened to GPTQ with grid range search.

The authors acknowledged some conceptual overlap but argued that LeanQuant introduces unique features, such as the use of inverse Hessian diagonal outliers, loss-error-aware grids, and scalability for models up to 405B parameters. The scalability argument is however weak as I don't think there is necessarily something wrong with other methods that prevent them to scale to larger models.

(b) (Reviewer evsw) noted the lack of direct comparisons with recent methods like QuaRot, QuIP, and SpinQuant in the main text.

The authors added additional experiments comparing LeanQuant with QuaRot, demonstrating that LeanQuant performs on par or better in accuracy while offering advantages in efficiency and scalability. They also noted that QuaRot introduces inference overheads that LeanQuant avoids, which is another contribution of this work.

(c) (Reviewer uLwi) raised the concern about the practical benefits of 2-bit and 3-bit quantization, given hardware limitations in supporting sub-4-bit operations. Reviewers questioned whether these lower bit widths could deliver computational benefits.

The authors argued that LLM inference is typically IO-bound rather than compute-bound, making memory savings more impactful than computational efficiency. They provided token throughput benchmarks showing significant speedups for 2-bit and 3-bit LeanQuant models. The reviewer did not engage in the post-rebuttal discussions and after reading the rebuttal I agree with the authors' response and found it satisfactory.

---

### Decision · Program_Chairs · 2025-01-22

Accept (Poster)